# Concept Algebra for (Score-Based) Text-Controlled Generative Models

Zihao Wang[1], Lin Gui[1], Jeffrey Negrea[2], and Victor Veitch[1,2,3]

[1]*Department of Statistics, University of Chicago*
[2]*Data Science Institute, University of Chicago*
[3]*Google Research*

## Abstract

This paper concerns the structure of learned representations in text-guided generative models, focusing on score-based models. A key property of such models is that they can compose disparate concepts in a 'disentangled' manner. This suggests these models have internal representations that encode concepts in a 'disentangled' manner. Here, we focus on the idea that concepts are encoded as subspaces of some representation space. We formalize what this means, show there's a natural choice for the representation, and develop a simple method for identifying the part of the representation corresponding to a given concept. In particular, this allows us to manipulate the concepts expressed by the model through algebraic manipulation of the representation. We demonstrate the idea with examples using Stable Diffusion.

## 1 Introduction

Large-scale text-controlled generative models are now dominant in many parts of modern machine learning and artificial intelligence [e.g., Bro+20; Rad+21; Bom+21; Koj+22]. In these models, the user provides a prompt in natural language and the model generates samples based on this prompt—e.g., in large language models the sample is a natural language response, and in text-to-image models the sample is an image. These models have a remarkable ability to compose disparate concepts to generate coherent samples that were not seen during training.This suggests that these models have some internal representation of high-level concepts that can be manipulated in a 'disentangled' manner. Broadly, the goal of this paper is to shed light on how this concept representation works, and how it can be manipulated. We focus on text-to-image diffusion models, though many of the ideas are generally applicable.

Our starting point is the following commonly observed structure of representations:

1. Each data point $x$ is mapped to some representation vector $\text{Rep}(x) \in \mathbb{R}^p$.

2. High-level concepts correspond to subspaces (directions) of the representation space.

Perhaps the best known example of this structure is in word embeddings, where semantic relationships such as $\text{Rep}(\text{"king"}) - \text{Rep}(\text{"queen"}) \approx \text{Rep}(\text{"man"}) - \text{Rep}(\text{"woman"})$ suggest that high-level concepts (here, sex) are encoded as directions in the representation space [Mik+13]. This kind of encoding of concepts has been argued to occur in many contexts, including in the latent space of variational autoencoders [ZW20; Khe+20; MFM21] and in the latent space of language models [Bol+16; GG19; Rad+21; Elh+22]. We'll call representations of this kind *arithmetically composable*, because composition corresponds to arithmetic operations on the representation vectors. The goal of this paper is to develop arithmetically composable representations of text for score-based text to image models.

There are two main motivations. First, understanding the structure of the representation space is important for foundational progress on understanding the emergent behavior of text-controlled generative models. It is particularly interesting to study this question in the text-to-image setting

37th Conference on Neural Information Processing Systems (NeurIPS 2023).

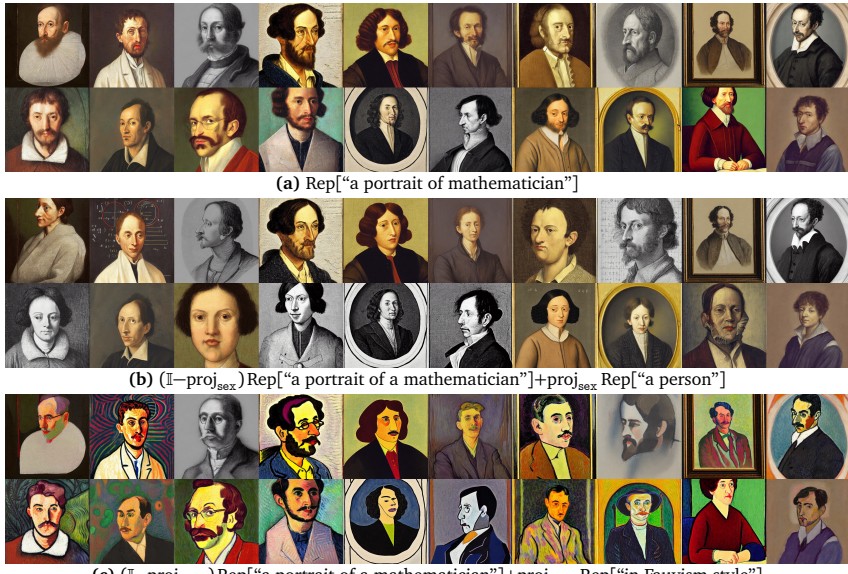

**(a)** Rep["a portrait of mathematician"]

**(b)** $(\mathbb{I}-\mathrm{proj}_{\mathrm{sex}})$ Rep["a portrait of a mathematician"]$+\mathrm{proj}_{\mathrm{sex}}$ Rep["a person"]

**(c)** $(\mathbb{I}-\mathrm{proj}_{\mathrm{style}})$ Rep["a portrait of a mathematician"]$+\mathrm{proj}_{\mathrm{style}}$ Rep["in Fauvism style"]

**Figure 1:** We show that high-level concepts such as sex and artistic_style are encoded as subspaces of a suitably chosen representation space. This allows us to manipulate the concepts expressed by a prompt through algebraic operations on the representation of that prompt. Namely, we edit the representation projected on to the subspace corresponding to a concept. Note images are paired by random seed.

because the multi-modality of the data makes it straightforward to distinguish concepts from inputs, and because it is not clear a priori that the models themselves build in any inductive bias towards arithmetic structure. The secondary motivation is that having such a representation would allow us to manipulate the concepts expressed by the model through linear-algebraic manipulations of representation of the input text; fig. 1 illustrates this idea.

The development of the paper is as follows:

1. We develop a mathematical formalism for describing the connection between representation structures and concepts for text-controlled generative models.

2. Using this formalism, we show that the Stein score of the text-conditional distribution is an arithmetically composable representation of the input text.

3. Then, we develop *concept algebra* as a method for manipulating the concepts expressed by the model through algebraic manipulation of this representation. We illustrate the approach with examples manipulating a variety of concepts.

## 2 A Mathematical Framework for Concepts as Subspaces

The first task is to develop a precise mathematical formalism for connecting the structure of representations and high-level concepts. This is necessary for understanding when such representations are possible, how to construct them, and when they may fail. We must make precise what is a concept, how concepts relate to inputs $x$, and what it means to represent concepts.

**Concepts** The real-world process that generated the training data has the following structure. First, images $Y$ are generated according to some real-world, physical process. Then, some human looks at each image and writes a caption describing it. Inverting this process, each text $x$ induces some probability density $p(y \mid x)$ over images $Y$ based on how compatible they are with $x$ as a caption. The (implicit) goal of the generative model is to learn this distribution.

To write the caption, the human first maps the image to a set of high-level variables summarizing the image's content, then uses these latent variables to generate the text $X$. Let $C$ be the latent variable that captures all the information about the image that is relevant for a human writing a caption. So,

$$p(y \mid X = x) = \int p(y \mid C = c)p(C = c \mid X = x)\mathrm{d}c. \tag{1}$$

The random variable $C$ captures the information that is jointly relevant for both the image and caption. Variables in $C$ include attributes such as has_mathematician or is_man, but not pixel_14_is_red. We define concepts in terms of the latent $C$.

**Definition 2.1.** A *concept variable $Z$* is a $C$-measurable random variable. The *concept $\mathscr{Z}$* associated to $Z$ is the sample space of $Z$.

Now, the full set of all possible concepts is unwieldy. Generally, we are concerned only with the concepts elicited by a particular prompt $x$.

**Definition 2.2.** A set of concepts $\mathscr{Z}_1, \ldots, \mathscr{Z}_k$ is *sufficient for $x$* if $p(c \mid X = x, Z_{1:k} = z_{1:k}) = p(c \mid Z_{1:k} = z_{1:k})$ for all $z_1, \ldots, z_k \in \mathscr{Z}_1 \times \cdots \times \mathscr{Z}_k$.

For example, the concept profession would be sufficient for the prompt "A nurse". This prompt induces a distribution on many concepts (e.g., background is likely to be a hospital) but these other concepts are independent of the caption given profession = nurse. Then,

$$p(y \mid x) = \sum_{z_{1:k}} p(y \mid z_{1:k}) p(z_{1:k} \mid x). \tag{2}$$

**Concept Distributions**  Following eq. (2), we can view each text $x$ as specifying a distribution $p(z_{1:k} \mid x)$ over latent concepts $\mathscr{Z}_1, \ldots, \mathscr{Z}_k$. This observation lets us make the relationship between text and concepts precise.

**Definition 2.3.** A *concept distribution $Q$* is a distribution over concepts. Each text $x$ specifies a concept distribution as $Q_x = p(z_{1:k} \mid x)$.

That is, we move from viewing text as expressing specific concept values (is_mathematician = 1) to expressing probability distributions over concepts ($Q_x$(is_mathematician = 1) = 0.99). The probabilistic view is more general—deterministically expressed concepts can be represented as degenerate distributions. This extra generality is necessary: for example, the prompt "a person" induces a non-degenerate distribution over the sex concept.

**Concept Representations**  A text-controlled generative model takes in prompt text $x$ and produces a random output $Y$. Implicitly, such models are maps from text strings $x$ to the space of probability densities over $Y$. We'll define a representation $\mathrm{Rep}(x) \in \mathscr{R}$ of $x$ as any function of $x$ that suffices to specify the output distribution. We define $f_r(\cdot)$ as the density defined by $r \in \mathscr{R}$, and assume that the model learn's the true data distribution of $Y|X = x$:

$$f_{\mathrm{Rep}(x)}(y) = p(y \mid x). \tag{3}$$

The key idea for connecting representations and concepts is to move from considering representations of prompts to representations of concept distributions.

**Definition 2.4.** A *concept representation* Rep is a function that maps a concept distribution $Q$ to a representation $\mathrm{Rep}(Q) \in \mathscr{R}$, where $\mathscr{R}$ is a vector space. The *representation of a prompt $x$* is the representation of the associated concept distribution, $\mathrm{Rep}(x) := \mathrm{Rep}(Q_x)$.

There are two reasons why this view is desirable. First, defining the representation in terms of the concept distribution makes the role of concepts explicit—this will allow us to explain how representation structure relates to concept structure.

The second reason is that it allows us to reason about representations that do not correspond to any prompt. Every prompt defines a concept distribution, but not every concept distribution can be defined by a prompt. This matters because we ultimately want to reason about the conceptual meaning of representation vectors created by algebraic operations on representations of prompts. Such vectors need not correspond to any prompt.

**Arithmetic Compositionality**  We now have the tools to define what it means for a representation to be arithmetically composable. We define composability for a pair of concepts $\mathscr{Z}$ and $\mathscr{W}$. In the subsequent development, our aim will be to manipulate $\mathscr{Z}$ while leaving $\mathscr{W}$ fixed.

**Definition 2.5.** A representation Rep is *arithmetically composable* with respect to concepts $\mathscr{Z}, \mathscr{W}$ if there are vector spaces $\mathscr{R}_Z$ and $\mathscr{R}_W$ such that for all concept distributions of the form $Q(z, w) = Q_Z(z) Q_W(w)$,

$$\mathrm{Rep}(Q_Z Q_W) = \mathrm{Rep}_Z(Q_Z) + \mathrm{Rep}_W(Q_W), \tag{4}$$

where $\text{Rep}_Z(Q_Z) \in \mathscr{R}_Z$ and $\text{Rep}_W(Q_W) \in \mathscr{R}_W$.

In words: we restrict to product distributions to capture the requirement that the concepts $\mathscr{Z}$ and $\mathscr{W}$ can be manipulated freely of each other (the typical case is that one or both of $Q_Z$ and $Q_W$ are degenerate, putting all their mass on a single point). Then, the definition requires that there are fixed subspaces corresponding to each concept in the sense that, e.g., changing only $Q_Z$ induces a change only in $\mathscr{R}_Z$.

## 3   The Score Representation

We now have an abstract definition of arithmetically composable representation. The next step is to find a specific representation function that satisfies the definition.

We will study the following choice.

**Definition 3.1.** The *score representation* $s[Q]$ of a concept distribution $Q$ is defined by:

$$s[Q](y) := \nabla_y \log \int p(y \mid z, w) Q(z, w) \mathrm{d}z \mathrm{d}w.$$

The *centered score representation* $\bar{s}[Q]$ is defined by $\bar{s}[Q] := s[Q] - s[Q_0]$.

Here, $s[Q]$ is itself a function of $y$ and the representation space $\mathscr{R}$ is a vector space of functions. This is a departure from the typical view of representations as elements of $\mathbb{R}^p$. The score representation can be thought of as a kind of non-parametric representation vector. The centered score representation just subtracts off the representation of some baseline distribution $Q_0$.[1]

The main motivation for studying the score representation is that

$$s[x](y) := s[Q_x](y) = \nabla_y \log p(y \mid x).$$

The importance of this observation is that $\nabla_y \log p(y \mid x)$ is learnable from data. In fact, this score function is ultimately the basis of many controlled generation models [e.g., HJA20; Ram+22; Sah+22], because it characterizes the conditional while avoiding the need to compute the normalizing constant [HD05; SE19]. Accordingly, we can readily compute the score representation of prompts in many generative models, without any extra model training.

**Causal Separability**   The score representation does not have arithmetically composable structure with respect to every pair of concepts. The crux of the issue is that concepts are reflected in the representation based on their effect on $Y$. If the way they affect $Y$ depends fundamentally on some interaction between two concepts, the representation cannot hope to disentangle them. Thus, we must rule out this case.

**Definition 3.2.** We say that $Y$ is *causally separable* with respect to $\mathscr{Z}, \mathscr{W}$ if there exist unique $Y$-measurable variables $Y_{\mathscr{Z}}$, $Y_{\mathscr{W}}$, and $\xi$ such that

1. $Y = g(Y_{\mathscr{Z}}, Y_{\mathscr{W}}, \xi)$ for some invertible and differentiable function $g$, and
2. $p(y_{\mathscr{Z}}, y_{\mathscr{W}}, \xi \mid z, w) = p(\xi) p(y_{\mathscr{Z}} \mid z) p(y_{\mathscr{W}} \mid w)$

Informally, the requirement is that we can separately generate $Y_{\mathscr{Z}}$ and $Y_{\mathscr{W}}$ as the part of the output affected by $\mathscr{Z}$ and $\mathscr{W}$ (and $\xi$ as the part of the image unrelated to $Z$ and $W$), then combine these parts to form the final image. That is, generating the visual features associated to a concept $\mathscr{W}$ can't require us to know the value of another concept $\mathscr{Z}$. As an example where causal separability fails, consider the concepts of species $\mathscr{W} = \{\text{deer}, \text{human}\}$ and sex $\mathscr{Z} = \{\text{male}, \text{female}\}$. It seems reasonable that there is a $Y$-measurable $Y_{\mathscr{W}}$ that is the species part of the image—e.g., the presence of fur vs skin, snouts vs noses, and so forth. However, there is no part of $Y$ that corresponds to a sex concept in a manner that's free of species. The reason is that the visual characteristics of sex are fundamentally different across species—e.g., male deer have antlers, but humans usually do not. In fig. 8 we test this example, finding that concept algebra fails in the absence of causal separability.

It turns out it suffices to rule out this case (all proofs in appendix):

---

[1]The representation space $\mathscr{R}$ is the same for all $Q_0$; the choice is arbitrary. We define $\bar{s}$ to ensure 0 is an element of $\mathscr{R}$. This is for theoretical convenience; we will see that only $s$ is required in practice.

**Proposition 3.1.** *If Y is causally separable with respect to $\mathcal{W}$ and $\mathcal{Z}$, then the centered score representation is arithmetically composable with respect to $\mathcal{W}$ and $\mathcal{Z}$.*

That is: the (centered) score representation is structured such that concepts correspond to subspaces of the representation space.

## 4 Concept Algebra

We have established that concepts correspond to subspaces of the representation space. We now consider how to manipulate concepts through algebraic operations on representations.

To modify a particular concept $\mathcal{Z}$ we want to modify the representation only on the subspace $\mathcal{R}_Z$ corresponding to $\mathcal{Z}$. For example, consider changing the style concept to Fauvism. Intuitively, we want an operation of the form:

$$s_{\text{edit}} \leftarrow (\mathbb{I}-\text{proj}_{\text{style}})s[\text{"a portrait of mathematician"}]+\text{proj}_{\text{style}}s[\text{"Fauvism style"}], \qquad (5)$$

where $\text{proj}_{\text{style}}$ is the projection onto the subspace corresponding to the style concept. The idea is that the representation of the original prompt $x_{\text{orig}}$ is unchanged except on the style subspace. On the style subspace, the representation takes on the value elicited by the new prompt $x_{\text{new}} = $ "Fauvism style".

There are two main challenges for putting this intuition into practice. First, because we are working with an infinite dimensional representation, it is unclear how to do the projection. Second, although we know that some $\mathcal{R}_Z$ exists, we still need a way to determine it explicitly.

### 4.1 Concept Manipulation through Projection

Following proposition 3.1, we have that

$$\bar{s}[Q_Z \times Q_W] = \bar{s}_Z[Q_Z]+\bar{s}_W[Q_W], \qquad (6)$$

for some representation functions $\bar{s}_Z$ and $\bar{s}_W$ with range in $\mathcal{R}_Z$ and $\mathcal{R}_W$ respectively. We have that the $Z$-representation space is

$$\mathcal{R}_Z = \text{span}(\{\bar{s}_Z[Q_Z] : Q_Z \text{ a concept distribution}\}). \qquad (7)$$

Our goal is to find a projection onto $\mathcal{R}_Z$.

The first obstacle is that $\mathcal{R}_Z$ is a function space, making algebraic operations difficult to define. The resolution is straightforward. In practice, score-based models generate samples by running a discretized (stochastic) differential equation forward in time. These algorithms only require the score function evaluated at the finite set of points. At each $y$, we have that $\bar{s}(y) \in \mathbb{R}^m$. Accordingly, by restricting attention to a single value of $y$ at a time, we can use ordinary linear algebra to define the manipulations:

**Definition 4.1.** The *Z subspace at $y$* is

$$\mathcal{R}_Z(y) := \text{span}(\{\bar{s}_Z[Q_Z](y) : Q_Z \text{ a concept distribution}\}) \qquad (8)$$

and the *Z-projection at $y$*, denoted $\text{proj}_Z(y)$ is the projection onto this subspace.

If we can compute $\text{proj}_Z(y)$ then we can just edit the representation at each point $y$. That is, we transform the score function at each point:

$$\bar{s}_{\text{edit}}(y) \leftarrow (\mathbb{I}-\text{proj}_Z(y))\bar{s}[x_{\text{orig}}](y)+\text{proj}_Z(y)\bar{s}[x_{\text{new}}](y). \qquad (9)$$

We then draw samples from the stochastic differential equation defined by $\bar{s}_{\text{edit}}$.

### 4.2 Identifying the Concept Subspace

The remaining obstacle is that we need to explicitly identify $\mathcal{R}_Z(y)$, so that we can compute $\text{proj}_Z(y)$. The problem is that the function $\bar{s}_Z$ in eq. (6) is unknown, so we cannot compute $\mathcal{R}_Z(y)$ directly. Our strategy for estimating the space is based on the following proposition.

**Proposition 4.1.** *Let $Q_W$ be any fixed distribution over the W concept and $Q_Z^0$ be any reference distribution over Z. Then, assuming causal separability for $\mathcal{Z}, \mathcal{W}$,*

$$\mathcal{R}_Z(y) = span(\{s[Q_Z Q_W](y)-s[Q_Z^0 Q_W](y) : Q_Z \text{ a concept distribution}\}). \qquad (10)$$

The importance of this expression is that it does not require the unknown $s_Z$.

We'll use eq. (10) to identify $\mathcal{R}_Z(y)$. The idea is to find a basis for the subspace using prompts $x_0, \ldots x_k$ that elicit distributions of the form $Q_x = Q_Z Q_W$. For example, to identify the sex concept we use the prompts $x_0 =$ "a man" and $x_1 =$ "a woman", with the idea that

$$Q_{x_0} = \delta_{\text{male}} \times Q_W \qquad\qquad Q_{x_1} = \delta_{\text{female}} \times Q_W, \tag{11}$$

with the same marginal distribution $Q_W$. We then use the prompts to define the estimated subspace as

$$\hat{\mathcal{R}}_Z(y) := \text{span}(\{s[x_i](y) - s[x_0](y) : i = 1, \ldots, k\}). \tag{12}$$

That is, we write $k+1$ prompts $x_0, \ldots x_k$ designed so that each elicits a different distribution over $Z$, but the same distribution on $W$. Then, the estimated subspace is given by eq. (12).

### 4.3 Concept Algebra

Summarizing, our approach to algebraically manipulating concepts is:

1. Find prompts $x_0, \ldots, x_k$ such that each elicits a different distribution on $Z$, but the same distribution on $W$. That is, $Q_{x_j} = Q_Z^j Q_W$ for each $j$.

2. Construct the estimated representation space $\hat{\mathcal{R}}_Z(y)$ following eq. (12), and define $\text{proj}_Z(y)$ as the projection onto this space.

3. Sample from the discretized SDE defined by the manipulated score representation[2]

$$s_{\text{edit}}(y) \leftarrow (\mathbb{I} - \text{proj}_Z(y)) s[x_{\text{orig}}](y) + \text{proj}_Z(y) s[x_{\text{new}}](y). \tag{13}$$

Implementation of eq. (13) with the diffusion model is described in appendix A.

## 5 Validity of Concept Subspace Identification

The procedure described in the previous section relies on finding spanning prompts $x_0, \ldots, x_k$ for the target concept subspace. These prompts must satisfy $Q_{x_j} = Q_Z^j Q_W$ for some common $Q_W$, and we must have sufficient prompts to span the subspace. The first condition is a question of prompt design, and is often not too hard in practice. However, it is natural to wonder when it's possible to actually recover $\mathcal{R}_Z$ using only a practical number of prompts. We give some results showing that the dimension of $\mathcal{R}_Z(y)$ is often small, and thus can be spanned with a small number of prompts. Note that these results rely on the special structure of the score representation, and may not hold for other representations.

First, the case where $\mathcal{Z}$ is categorical with few categories:

**Proposition 5.1.** *Assuming causal separability holds for $\mathcal{Z}, \mathcal{W}$. If $\mathcal{Z}$ is categorical with $L$ possible values ($L \geq 2$), then $dim(\mathcal{R}_Z(y)) \leq L - 1$.*

This result covers concepts such as sex, which can be spanned with only two prompts.

The next result extends this to certain categorical concepts with large cardinality, such as style. The idea is that if a concept is composed of finer grained categorical concepts, each with small cardinality, then the representation space of the concept is also low-dimensional. For example, style may be composed of lower-level concepts such as color, stroke, textures, etc.

**Proposition 5.2.** *Suppose $\mathcal{Z}$ is composed of categorical concepts $\{\mathcal{Z}_k\}_{k=1}^K$ each with the number of categories $L_k$, in the sense that $\mathcal{Z} = \mathcal{Z}_1 \times \ldots \mathcal{Z}_k$. Assume $Y$ satisfies causal separability with respect to $\mathcal{Z}, \mathcal{W}$, with $Y_\mathcal{Z}$ the corresponding $Y$-measurable variable for $\mathcal{Z}$. Further assume that there exists $Y_\mathcal{Z}$-measurable variables $Y_{\mathcal{Z}_k}$ such that $p(y_\mathcal{Z} \mid z) = \Pi_{k=1}^K p(y_{\mathcal{Z}_k} \mid z_k)$. Then*

$$dim(\mathcal{R}_Z(y)) \leq \sum_{k=1}^K (L_k - 1) \tag{14}$$

---

[2] We can view this as first editing the centered representation $\bar{s}$: $\bar{s}_{\text{edit}}(y) \leftarrow (\mathbb{I} - \text{proj}_Z(y)) \bar{s}[x_0](y) + \text{proj}_Z(y) \bar{s}[\tilde{x}](y)$. Then add the same baseline on both sides.

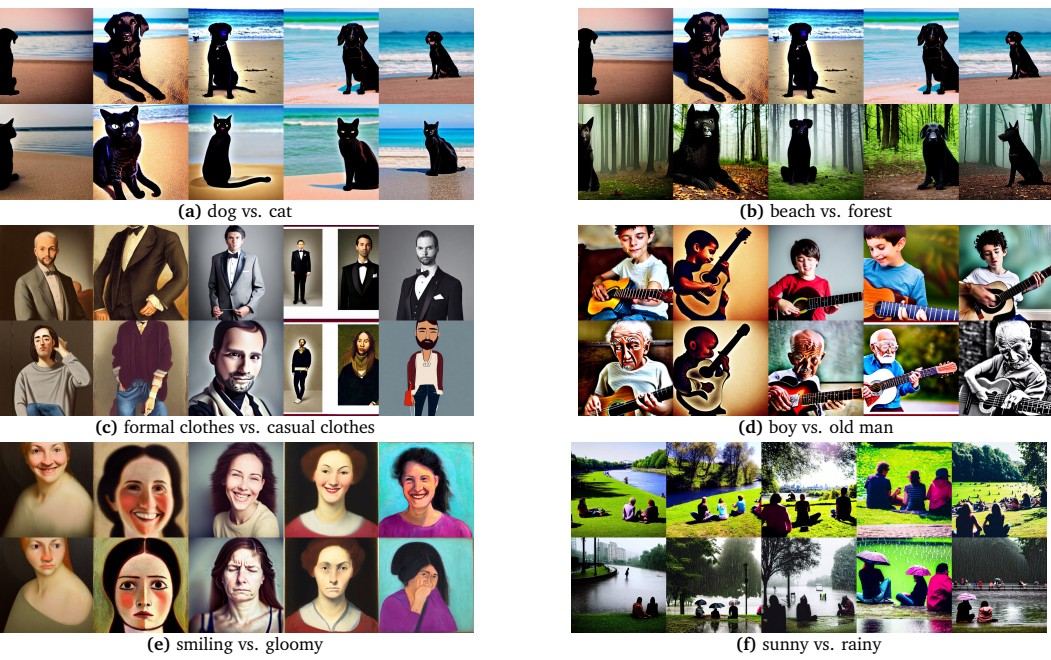

**(a)** dog vs. cat  **(b)** beach vs. forest

**(c)** formal clothes vs. casual clothes  **(d)** boy vs. old man

**(e)** smiling vs. gloomy  **(f)** sunny vs. rainy

**Figure 2:** Binary concepts (e.g. dog−vs−cat, smiling−vs−gloomy) correspond to subspaces, and can be easily manipulated with concept algebra.

Following this result, we might take the spanning prompts for style to be $x_0$ = "a mathematician in Art Deco style", $x_1$ = "a mathematician in Impressionist style", etc. Each of these prompts elicit a fixed distribution $Q_W$ over the content, but varies the distribution $Q_Z$ over style. If style is composed of finer-grained attributes, a relatively small set of such of prompts will suffice.

## 6 Experiments

We have formalized what it means for concepts to correspond to subspaces of the representation space, and derived a procedure for identifying and editing the subspaces corresponding to particular concepts in the score representation. We now work through some examples testing if this subspace structure does indeed exist, and whether it enables manipulation of concepts.

**Many concepts are represented as subspaces** First, we check whether the subspace structure does indeed exist. To this end, we generate randomly selected prompts—e.g., "a black dog sitting on the beach"—and attempt to change binary concepts expressed in the prompt. For example, we change the subject to be a cat by manipulating the concept dog−vs−cat with concept algebra. This, and other examples, are shown in fig. 2. It is clear that we are able to manipulate the target concept—providing evidence that these concepts are represented as subspaces.

**Concept Algebra can disentangle hard-to-separate concepts** We stress-test concept algebra by using it to sample images expressing combinations of concepts that occur rarely in the training data. Specifically, we look at unlikely subject/style combinations—e.g., "A nuclear power plant in Baroque painting". We accomplish this by taking a base prompt that generates a high-quality image, but with the wrong style (e.g., "A nuclear power plant"). Then we use concept algebra to edit the style. We compare this with directly prompting the model, and with concept composition [e.g., Du+21; Liu+21]. The later is a method that adds on a score representation of the style to the base prompt (without projecting on to a target subspace).

We used each of the three methods to sample images expressing to 49 anti-correlated content/style pairs. Human raters were then presented with the outcomes alongside reference images, and asked to rank them based on adherence to the desired style and content. This evaluation was replicated across 10 different raters. Refer to fig. 3 for illustrative examples. Raters consistently favored images produced by concept algebra, as highlighted in table 1. This aligns with our theory, suggesting concept algebra's adeptness in retaining content while altering style. See appendix C for further details.

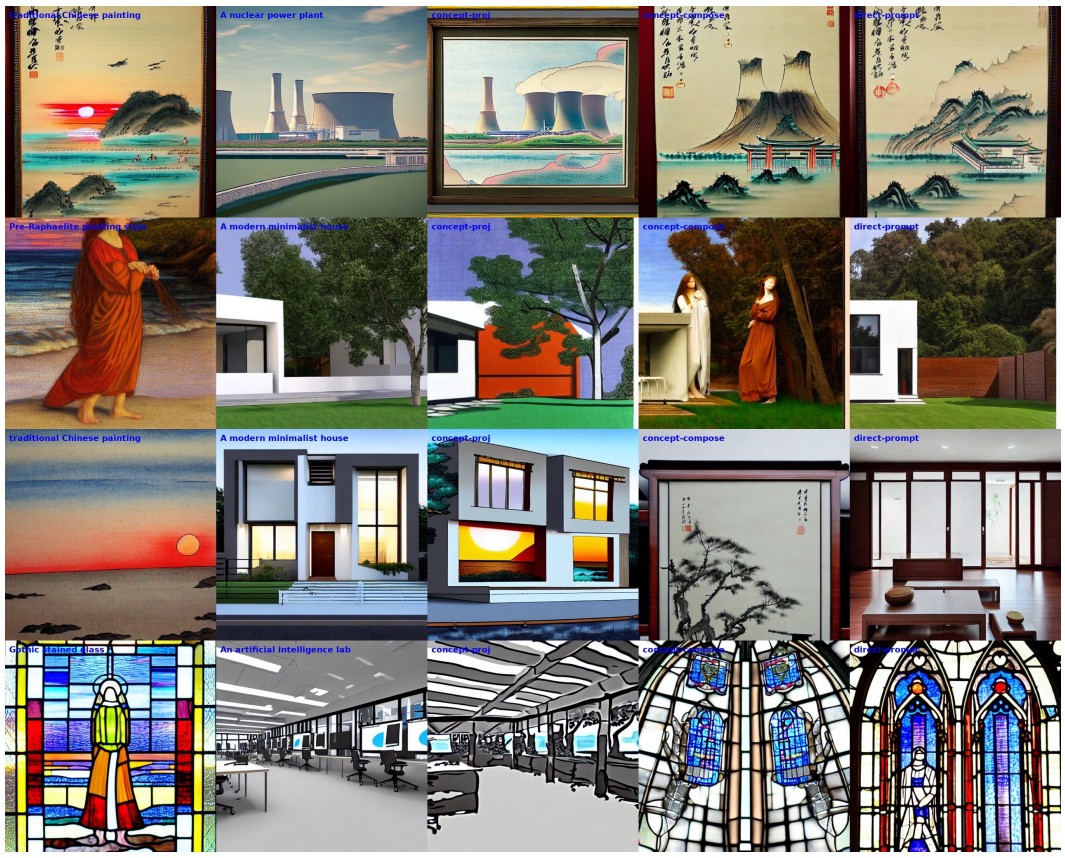

**Figure 3:** Concept algebra is considered more faithful than other methods. Raters are shown rows of images and rank method outputs by how well they produce target style while preserving content. Each row has style-reference (leftmost), a content-reference (2nd from left), and left-to-right (here, but randomly permuted in the survey) images generated from concept-algebra, composition and direct prompting. Quantative results are in table 1.

|                    | Direct Prompting | Concept Composition | Concept Algebra | All Bad |
|--------------------|------------------|---------------------|-----------------|---------|
| Average Proportion | 0.162            | 0.164               | **0.476**       | 0.198   |
| Standard Error     | 0.017            | 0.017               | 0.023           | 0.018   |

**Table 1:** Raters find concept algebra is more faithful to content and style than direct prompting or concept composition

**Mixture concept distributions** Next, our theory predicts that we can use concept algebra to sample from mixture (non-degenerate) distributions over concept values. Consider sex. Figure 1a shows that $x =$ "a portrait of a mathematician" almost always generates pictures of men. In fig. 1b we sample from the distribution induced by

$$s_{\text{edit}} \leftarrow (\mathbb{I}-\text{proj}_{\text{sex}})s[\text{x}]+\text{proj}_{\text{sex}}s[\text{"person"}], \tag{15}$$

and observe that we indeed get a mixture distribution (induced by "person") over sex.

**Non-prompted edits to the subspace edit the concept** Concept algebra uses reference prompts (e.g., "woman" or "person") to set the target concept distribution. It's natural to ask what happens if we make an edit to a concept subspace that does not correspond to a reference prompt. In fig. 4, we sample from

$$s_{\text{edit}} \leftarrow (\mathbb{I}-\text{proj}_{\text{sex}})s[\text{x}]+\text{proj}_{\text{sex}}(\frac{1}{2}s[\text{"a male nurse"}]+\frac{1}{2}s[\text{"a female nurse"}]). \tag{16}$$

The representation vector $\frac{1}{2}s[\text{"a male nurse"}]+\frac{1}{2}s[\text{"a female nurse"}]$ need not correspond to any English prompt. We observe that modifications on the subspace still affect just the `sex` concept though—the samples are androgynous figures!

**Mask as a Concept**  Finally, we consider a more abstract kind of concept motivated by the following problem. Suppose we have several photographs of a particular toy, and we want to generate an image of this toy in front of the Eiffel Tower. In principle, we can do this by fine-tuning the model (e.g., with dreambooth) to associate a new token (e.g., "sks toy") with the toy. Then, we can generate the image by conditioning on the prompt "a sks toy in front of the Eiffel Tower". In practice, however, this can be difficult because the fine-tuning ends up conflating the toy with the background in the demonstration images. E.g., the prompt "a sks toy in front of the Eiffel Tower" tends to generate images featuring carpet; see fig. 5b.

Intuitively, we might hope to fix this problem by finding a concept subspace that excludes background information. Given such a "subject subspace", we could mask the subject out of the image, generate the background, and then edit the subject back in. In appendix C.1 we explain how to construct such a subspace using the prompts $x_0 = $ "a toy" and $x_1 = $ "a soccer ball". Figure 5 shows the sampled output.

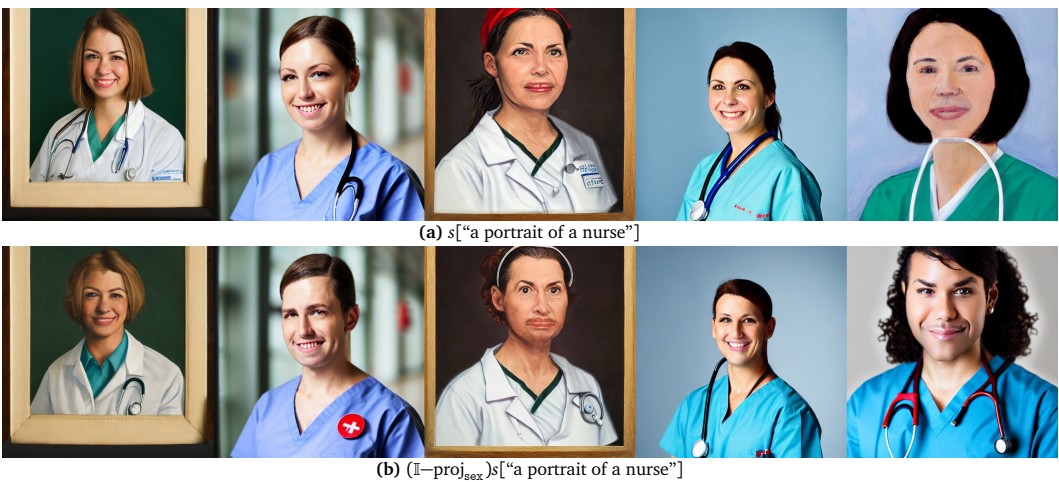

**(a)** $s[\text{"a portrait of a nurse"}]$

**(b)** $(\mathbb{I}-\text{proj}_{\text{sex}})s[\text{"a portrait of a nurse"}]$
$+\text{proj}_{\text{sex}}(\frac{1}{2}s[\text{"a female nurse"}]+\frac{1}{2}s[\text{"a male nurse"}])$

**Figure 4:** Elements of the $\mathscr{R}_Z$ may not correspond to any prompt.

## 7   Discussion and Related Work

We introduced a framework illustrating that concepts align with subspaces of a representation space. Through this, we validated the structure of the score representation and derived a method to identify the subspace for a concept. We then demonstrated concept manipulation in a diffusion model's score representation.

**Concepts as Subspaces**  There has been significant interest in whether and how neural representations encode high-level concepts. A substantial body of work suggests that concepts correspond to subspaces of a representation space [e.g., Mik+13; MYZ13; PSM14; GL14; Aro+15; GAM17; AH19]. Often, this work focuses on a specific representation learning approach and is either empirical or offers domain-tied theoretical analysis. For instance, in word embeddings, theories explaining observed structures depend on the unique nature of language [e.g., Aro+15; AH19]. In contrast, our paper's mathematical development is broad—we merely stipulate that the data have two views separated by a semantically meaningful space. We argue that the concepts-as-subspaces structure stems from the structure of probability theory, independent of any specific architecture or algorithm.

Our work also relates to studies that assume training data arises from a specific latent variable model and demonstrate that learned representations (partially) uncover these latent variables [e.g., HM16; HM17; HST19; Khe+20; VK+21; Eas+22; Hig+18; Zim+21]. This literature often aims for "disentangled" representations where each latent space dimension matches a single latent factor. Unlike these, we don't presuppose a finite set of latent factors driving the data.

Instead, our representations define probability distributions over latent concepts, not merely recovering them. This non-determinism, as observed, is generally necessary.

**Controlling Diffusion Models**   To demonstrate the concept-as-subspace structure, we developed a method for identifying the subspace corresponding to a given concept and showed how to manipulate concepts in the score representation of a diffusion model. We emphasize that our contribution here is not the manipulation procedure itself, but rather the mathematical framework that makes this procedure possible. In particular, the requirement to manipulate entire score functions is somewhat burdensome computationally. However, the ability to precisely manipulate individual concepts is clearly a useful tool, and it is an intriguing direction for future work to develop more efficient procedures for doing so. We conclude by surveying connections to existing work on controlling diffusion models.

One idea has been to take the bottleneck layer of UNet as a representation space and control the model by manipulating this space [KJU22; Haa+23; Par+23]. This work does not consider text controlled models. It would be intriguing to understand the connection to the score-representation view, as moving from manipulation of the score to manipulation of the bottleneck layer would be a large computational saving.

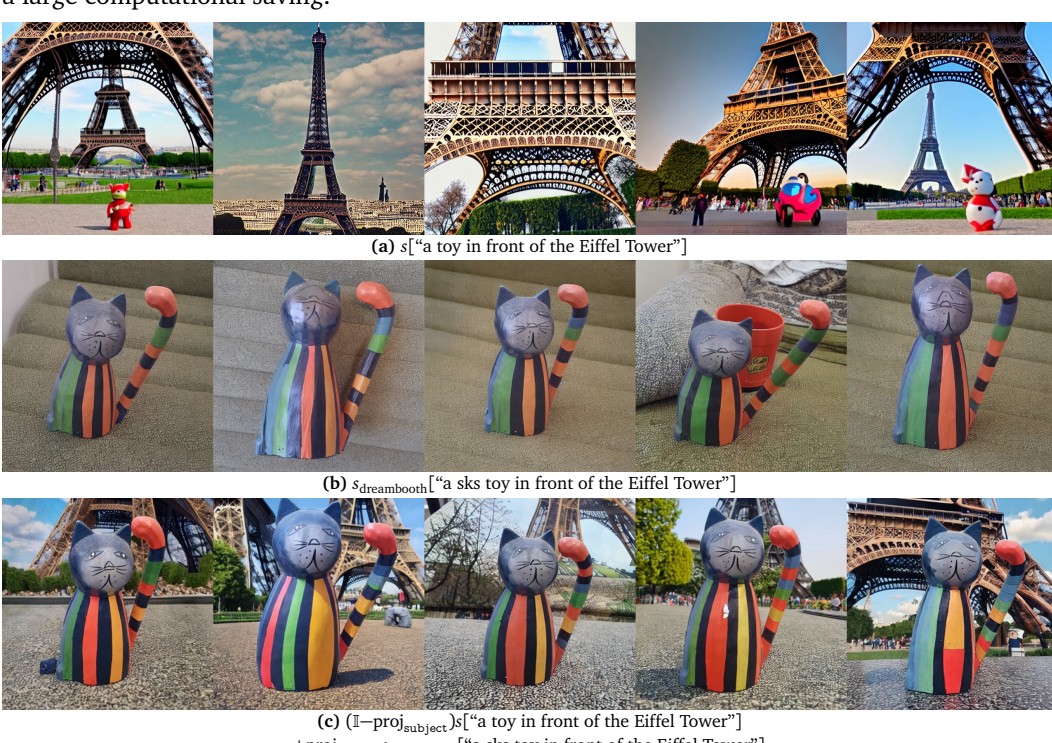

**(a)** $s[$"a toy in front of the Eiffel Tower"$]$

**(b)** $s_{\text{dreambooth}}[$"a sks toy in front of the Eiffel Tower"$]$

**(c)** $(\mathbb{I}-\text{proj}_{\text{subject}})s[$"a toy in front of the Eiffel Tower"$]$
$+\text{proj}_{\text{subject}}s_{\text{dreambooth}}[$"a sks toy in front of the Eiffel Tower"$]$

**Figure 5:** We can manipulate abstract concepts such as 'subject' of the image

Concept algebra can be seen as providing a unifying mathematical view on several methods that manipulate the score function [e.g., Du+21; Liu+21; NBP22; Ano23]. Du et al. [DLM20] and Liu et al. [Liu+22] manipulate concepts via adding and subtracting scores. Negative prompting is a widely-used engineering trick that 'subtracts off' a prompt expressing unwanted concepts. In section 6 and appendix D we compared against these heuristics and show that concept algebra is more effective at manipulating concepts in isolation. Couairon et al. [Cou+22] use score differences to identify objects' locations in images; this inspired our approach in section 6 and appendix C.1. In each case, we have seen that this kind of manipulation may be viewed as editing the subspace corresponding to some concept.

## Acknowledgements

This work is supported by ONR grant N00014-23-1-2591 and Open Philanthropy.

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

# A Concept Algebra Algorithms in Diffusion Model

Text-to-image Diffusion Models use score representations in their generation. More specifically, suppose the target is to sample $Y = Y_0 \sim P^*$, with the corresponding score function denoted as $s_0$. The key ingredients for generation are the score function for $Y_t$ (denoted as $s_t$), which is $Y$ noised at different levels, (e.g. $Y_t = (1-\alpha_t)Y + \alpha_t \epsilon_t$ for standard independent Gaussian noise $\epsilon$), for $t = 0, ..., T$. See [Luo22] for more details. To apply our results, we require causal separability with respect to $\mathscr{Z}, \mathscr{W}$ holds for all $Y_t, t = 0, ..., T$. Then our theoretical results follow through.

Algorithm 1 is an implementation of Concept Manipulation through Projection based on DDPM [HJA20] (we can also implement different variants). [3] It requires FindSubspaceMethod, for which we can use FindSubspaceBasis(algorithm 2) and FindSubspaceMask(algorithm 3) based on the properties of $\mathscr{Z}$ as discussed in the main text. More specifically,

**FindSubspaceBasis** We calculate the projection matrix (denoted as $\Pi_Z$) for the $Z$-subspace, from a span of $K$ prompts (after subtracting off the baseline) (algorithm 2). In practical computations, we evaluate the $m \times K$ matrix $\triangle E$:

$$\triangle E := [\epsilon_\theta(y_t, t \mid x_1) - \epsilon_\theta(y_t, t \mid x_0), ..., \epsilon_\theta(y_t, t \mid x_K) - \epsilon_\theta(y_t, t \mid x_0)]$$

Then, the top $K_{\text{thres}}$ left singular vectors are selected as $Q$. Here, $K_{\text{thres}}$ denotes the least number of factors required to surpass a certain proportion of variance explained, denoted as thres. Consequently, we have $\Pi_z \leftarrow QQ^T$.

**FindSubspaceMask** In this context, $\Pi_Z$ signifies a mask. This mask can be calculated from the score difference $\triangle \epsilon$ (refer to algorithm 3). As a practical measure, we may implement noise reduction techniques to fine-tune $\triangle \epsilon$. One approach is the application of a Gaussian blur to smooth out neighboring pixels.

---

**Algorithm 1** Concept Manipulation through Projection

---

1: **Require** Diffusion model $\epsilon_\theta(y_t, t \mid x)$, guidance scale $w$, covariance matrix $\sigma_t^2 I$,
   empty prompt "", prompts: $x_{\text{orig}}, x_{\text{new}}$,
   prompts to build the $Z$ subspace: $\{x_i\}$,
   the function for finding the $Z$ subspace: FindSubspaceMethod$(\cdot, \cdot)$
2: Initialize sample $y_T \sim \mathcal{N}(\mathbf{0}, \mathbf{I})$
3: **for** $t = T, \ldots, 1$ **do**
4:     $\epsilon_{\text{empty}} \leftarrow \epsilon_\theta(y_t, t \mid \text{""})$ # unconditional score
5:     $\epsilon_{\text{orig}}, \epsilon_{\text{new}} \leftarrow \epsilon_\theta(y_t, t \mid x_{\text{orig}}), \epsilon_\theta(y_t, t \mid x_{\text{new}})$ # conditional scores
6:     $\Pi_Z \leftarrow$ FindSubspaceMethod$(y_t, \{x_i\})$ # find the projection matrix
7:     $\epsilon_{\text{cond}} \leftarrow (\mathbb{I} - \Pi_Z)\epsilon_{\text{orig}} + \Pi_Z \epsilon_{\text{new}}$ # concept projection
8:     $\epsilon \leftarrow \epsilon_0 + w(\epsilon_{\text{cond}} - \epsilon_0)$ # apply classifier-free guidance
9:     $y_{t-1} \sim \mathcal{N}\left(y_t - \epsilon, \sigma_t^2 I\right)$
10: **end for**

---

**Algorithm 2** FindSubspaceBasis

---

**Require:** $y_t \in \mathbb{R}^m$, prompts $\{x_k\}_{k=0}^K$
1: $\hat{\mathscr{R}}_Z(y) \leftarrow \text{span}(\{\epsilon_\theta(y_t, t \mid x_k) - \epsilon_\theta(y_t, t \mid x_0)\}_{k=1}^K)$
2: Determine $\Pi_Z$ as the projection matrix onto $\hat{\mathscr{R}}_Z(y)$
3: **return** $\Pi_Z$

---

[3]Note there here we use residual $\epsilon_\theta(y_t, t \mid x)$ instead of the score $s_\theta(y_t, t \mid x)$ for generation, they are equivalent up to a time-varying constant.

**Algorithm 3** FindSubspaceMask

---

**Require:** $y_t \in \mathbb{R}^m$, a pair of prompts $(x_1, x_2)$
1: $\triangle \epsilon \leftarrow \epsilon_\theta(y_t, t \mid x_1) - \epsilon_\theta(y_t, t \mid x_2)$
2: **for** $i = 1$ to $m$ **do**
3:     $m_i \leftarrow \triangle \epsilon_i \neq 0?1:0$
4: **end for**
5: $\Pi_Z \leftarrow \text{diag}(m_1, m_2, \ldots, m_m)$
6: **return** $\Pi_Z$

---

## B   Proofs

**Proposition 3.1.** *If $Y$ is causally separable with respect to $\mathcal{W}$ and $\mathcal{Z}$, then the centered score representation is arithmetically composable with respect to $\mathcal{W}$ and $\mathcal{Z}$.*

*Proof.* By assumption in Theorem 3.2, we have

$$p(y \mid z, w) = p(y_{\mathcal{Z}}, y_{\mathcal{W}}, \xi(y) \mid z, w) \left| \det\left( \frac{\partial g}{\partial y} \right) \right|$$

$$= p(y_{\mathcal{Z}} \mid z) p(y_{\mathcal{W}} \mid w) p(\xi(y)) \left| \det\left( \frac{\partial g}{\partial y} \right) \right|$$

Therefore,

$$p[Q](y) = p[Q_Z \times Q_W](y)$$

$$= p_Z[Q_Z](y) p[Q_W](y) p(\xi(y)) \left| \det(\frac{\partial g}{\partial y}) \right|,$$

where $p_Z[Q_Z](y) = \int p(y_{\mathcal{Z}} \mid z) Q_Z(z) \mathrm{d}z$ and $p_W[Q_W](y) = \int p(y_{\mathcal{W}} \mid z) Q_W(w) \mathrm{d}w$.

Then, taking the log-derivative with respect to $y$, we get its score function as follows:

$$s[Q_Z \times Q_W](y) = s_Z[Q_Z](y) + s_W[Q_W](y) + s_0(y) \tag{17}$$

where $s_Z(y)$ and $s_W(y)$ are $p_Z[Q_Z](y)$'s and $p_W[Q_W](y)$'s score functions, and $s_0(y) := \nabla_y \log\left( p(\xi(y)) \left| \det(\frac{\partial g}{\partial y}) \right| \right)$. So the centered-score is

$$\bar{s}[Q_Z \times Q_W](y) = (s_Z[Q_Z](y) - s_Z[Q_Z^0](y)) + (s_W[Q_W](y) - s_W[Q_W^0](y)) \tag{18}$$

where $Q_Z^0$ and $Q_W^0$ are the marginal distributions of $Z$ and $W$ of the baseline $Q_0$. Then, we can use the fact that

$$\mathcal{R}_Z = \text{span}(\{\bar{s}_Z[Q_Z] - \bar{s}_Z[Q_Z^0] : Q_Z \text{ a concept distribution}\})$$

$$= \text{span}(\{s_Z[Q_Z] - s_Z[Q_Z^0] : Q_Z \text{ a concept distribution}\})$$

$$\mathcal{R}_W = \text{span}(\{\bar{s}_W[Q_W] - \bar{s}_W[Q_W^0] : Q_W \text{ a concept distribution}\})$$

$$= \text{span}(\{s_W[Q_W] - s_W[Q_W^0] : Q_W \text{ a concept distribution}\})$$

Consequently, the claim follows. $\qquad\square$

**Proposition 4.1.** *Let $Q_W$ be any fixed distribution over the $W$ concept and $Q_Z^0$ be any reference distribution over $Z$. Then, assuming causal separability for $\mathcal{Z}, \mathcal{W}$,*

$$\mathcal{R}_Z(y) = span(\{s[Q_Z Q_W](y) - s[Q_Z^0 Q_W](y) : Q_Z \text{ a concept distribution}\}). \tag{10}$$

*Proof.* By causal separability we can easily get the $\mathcal{R}_Z(y)$ in proposition 4.1 is the same as:

$$\mathcal{R}_Z(y) = \text{span}(\{s_Z[Q_Z](y) - s_Z[Q_Z^0](y)\} : Q_Z \text{ a concept distribution})$$

The only thing left to show is that $\mathcal{R}_Z(y)$ remains the same for whatever choice of baseline $Q_Z^0$. But this is immediate: $\text{span}(\{s_Z[Q_Z](y) - s_Z[Q_Z^0](y) : Q_Z \text{ a concept distribution}\}) = \text{span}(\{s_Z[Q_Z](y) - s_Z[Q_Z^1](y) : Q_Z \text{ a concept distribution}\})$ for any two baselines $Q_Z^0$ and $Q_Z^1$. $\quad\square$

**Proposition 5.1.** *Assuming causal separability holds for $\mathscr{X}, \mathscr{W}$. If $\mathscr{X}$ is categorical with $L$ possible values ($L \geq 2$), then $dim(\mathscr{R}_Z(y)) \leq L{-}1$.*

*Proof.* We denote the possible values that $\mathscr{X}$ can take as $\{z_0, z_1, \ldots, z_{L-1}\}$. Let $\delta_{z_i} := \delta_{z_i}(z)$ represent the delta function in the $\mathscr{X}$-subspace, which is infinite at $z_i$ and zero at all other points. For any distribution $Q_Z$ over $Z$ and any $y \in \mathbb{R}^m$, we can express $s_Z[Q_Z](y)$ as a linear combination of $s_z[\delta_{z_i}]$ in the following form:

$$s_Z[Q_Z](y) = \sum_{l=0}^{L-1} \pi_l(y) s_z[\delta_{z_l}](y)$$

Here, $\sum_{l=0}^{L-1} \pi_l(y) = 1$. Consider a baseline concept distribution $Q_Z^0$ and its corresponding $Z$-related score $s_Z[Q_Z^0](y) = \sum_{l=0}^{L-1} c_l(y) s_z[\delta_{z_l}](y)$. We can then express the difference $s_Z[Q_Z](y) - s_Z[Q_Z^0](y)$ as:

$$s_Z[Q_Z](y) - s_Z[Q_Z^0](y) = \sum_{l=1}^{L-1} \omega_l(y)(s_z[\delta_{z_l}](y) - s_z[\delta_{z_0}](y)),$$

where $\omega_l(y) = \pi_l(y) - c_l(y)$ for $l = 1, \ldots, L-1$. Consequently, we can observe that $\mathscr{R}_Z(y) \subset \text{span}(\{s_z[\delta_{z_l}](y) - s_z[\delta_{z_0}](y)\}_{l=1}^{L-1})$, which implies that $dim(\mathscr{R}_Z(y)) \leq L{-}1$. $\qquad\square$

**Proposition 5.2.** *Suppose $\mathscr{X}$ is composed of categorical concepts $\{\mathscr{X}_k\}_{k=1}^K$ each with the number of categories $L_k$, in the sense that $\mathscr{X} = \mathscr{X}_1 \times \ldots \mathscr{X}_k$. Assume $Y$ satisfies causal separability with respect to $\mathscr{X}, \mathscr{W}$, with $Y_{\mathscr{X}}$ the corresponding Y-measurable variable for $\mathscr{X}$. Further assume that there exists $Y_{\mathscr{X}}$-measurable variables $Y_{\mathscr{X}_k}$ such that $p(y_{\mathscr{X}} \mid z) = \Pi_{k=1}^K p(y_{\mathscr{X}_k} \mid z_k)$. Then*

$$dim(\mathscr{R}_Z(y)) \leq \sum_{k=1}^K (L_k{-}1) \tag{14}$$

*Proof.* By assuming that $p(y_{\mathscr{X}} \mid z) = \prod_{k=1}^K p(y_{\mathscr{X}_k} \mid z_k)$, we can easily derive the following result for any concept distribution $Q_Z$ over $Z$:

$$s_Z[Q_Z](y) = \sum_{k=1}^K s_{Z_k}[Q_{Z_k}](y),$$

where $Q_{Z_k}$ represents the concept distribution of $\mathscr{X}_k$ for each $k$, and $s_{Z_k}[Q_{Z_k}](y) = \nabla_y \log\left(\int p(y_{\mathscr{X}_k} \mid z_k) Q_{Z_k}(z_k) dz_k\right)$. Recall that

$$\mathscr{R}_Z(y) = \text{span}(\{s_Z[Q_Z](y) - s_Z[Q_Z^0](y)\} : Q_Z \text{ is a concept distribution}),$$

where $Q_Z^0$ is a baseline. Importantly, it should be noted that $\mathscr{R}_Z(y)$ is unique regardless of the choice of $Q_Z^0$ as per [proposition 4.1](#).

Let $Q_{Z_k}^0$ denote the $\mathscr{X}_k$-related part of $Q_Z^0$ for $k = 1, \ldots, K$. We define $\mathscr{R}_{Z_k}(y) := \text{span}(\{s_{Z_k}[Q_{Z_k}](y) - s_{Z_k}[Q_{Z_k}^0](y)\})$. Then, we can state that:

$$\mathscr{R}_Z(y) \subset \sum_{k=1}^K \mathscr{R}_{Z_k}(y).$$

Based on [proposition 5.1](#), it follows that $dim(\mathscr{R}_{Z_k}(y)) \leq L_k{-}1$ for each $k$. Hence, we can conclude that:

$$dim(\mathscr{R}_Z(y)) \leq \sum_{k=1}^K (L_k{-}1).$$

$\qquad\square$

## C Experiment Details and More Figures

### C.1 Concept projection for Dreambooth (Figure 5)

First, we fine-tune the diffusion model using Dreambooth, applying a learning rate of $5e^{-6}$ and setting the number of steps to 800. While there are configurations that could yield a less overfitted model, we intentionally opt for these parameters to generate an overfitted model. Our aim is to verify if it's possible to disentangle the overfitted model by using concept manipulation via projection.

To generate images depicting a sks toy in front of the Eiffel Tower, we utilize our Dreambooth fine-tuned diffusion model together with the original pretrained Stable Diffusion model. Only for the new prompt, $x_{new} =$ a sks toy", we use the score function from the Dreambooth fine-tuned model. All other prompts are plugged into the score functions from the original pretrained Stable Diffusion model. To create the desired images, we construct a projector using a pair of prompts: $(x_1, x_2) = $ ("a toy", "a soccer ball"). The mask, computed using algorithm 3 with the threshold$= 0.1$, helps identify specific areas corresponding to the location of the subject. Then, we use the Dreambooth score function $s_{dreambooth}$(a sks toy") to guide the generation process within the masked region (areas with value 1), while using $s$(a toy in front of the Eiffel Tower") to guide the generation outside the mask (areas with value 0).

To ensure image fidelity, we exclusively employ the score function $s_{dreambooth}$("a sks toy") for guiding the denoising process for the last 6% of the denoising steps.

It is important to note that due to severe overfitting issues with the fine-tuned model, there is no significant difference between using either the prompt "a sks toy" or "a sks toy in front of the Eiffel Tower" for the fine-tuned model. Also, due to the same reason, we apply the original pretrained diffusion model for all score functions except for the sks toy related one.

### C.2 The mathematician example (Figure 1)

Our starting point is an original prompt $x_{orig} = $ "a portrait of a mathematician". Our objective is to modify the sex and style using concept projection:

To adjust sex, we formulate a corresponding direction using a pair of prompts $(x_1, x_2) = $ ("a man", "a woman"). Subsequently, we set $x_{new} = $ "a person".

To alter the style, we set $x_{new} = $ "a portrait of a mathematician, in Fauvism style". We define the concept subspace using prompts of the form "a portrait of a mathematician in $[x_{style}]$ style", where $x_{style}$ takes value from a list of styles. During sampling, the original prompt is utilized in the first 20% of timesteps to better retain the content.

The list of styles is generated by ChatGPT. They are: Art Deco, Minimalist, Baroque, Abstract Expressionist, Cubist, Fauvism, Impressionist, Steampunk, Neoclassical, Japanese Ukiyo-e, Surrealism, Memphis Design, Scandinavian, Bauhaus, Pop Art, Art Nouveau, Street Art, American West, Victorian Gothic, Futurism, Photorealistic, Mannerist, Flemish, Byzantine, Medieval, Romanesque, Trompe-l'œil, and Dutch Golden Age.

### C.3 Stress-test experiments (Figure 3)

In this experiemnt, we deliberately chose a diverse array of artistic styles and contrasting content to challenge our model. The styles used are Baroque painting, traditional Chinese painting, Pop art, Gothic stained glass, the Pre-Raphaelite painting style, Victorian botanical illustration, and Japanese Edo period art. In terms of content, we use: a bustling train station, a nuclear power plant, an artificial intelligence lab, a jazz music concert, a modern dance festival, a modern minimalist house, and a contemporary office setting. The rationale behind this selection was to pair styles and content that would rarely co-occur in training data. This rarity poses a significant challenge for the model in generating realistic outputs, testing its capabilities and adaptability to unconventional combinations.

We implement concept algebra the same way as in Figure 1 example. For concept composition, we use the software in [Liu+22].

In fig. 6 we show some examples where concept algebra is not preferred.

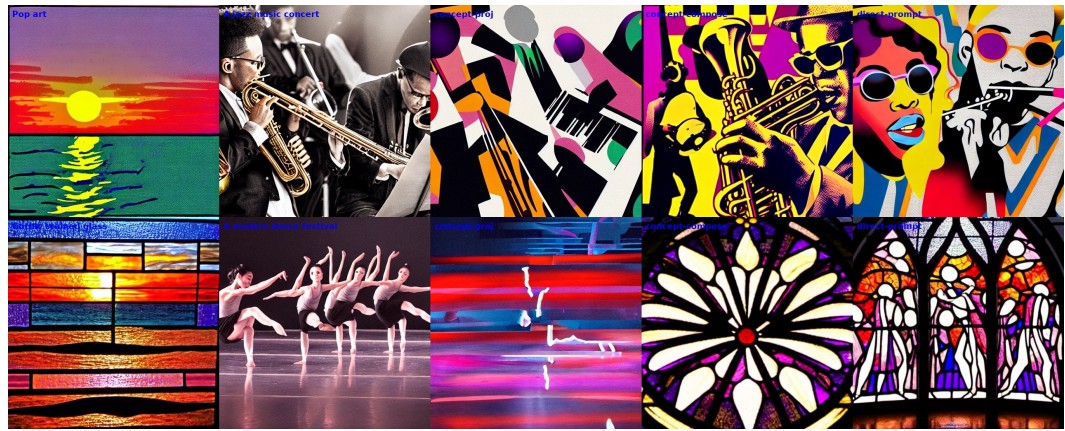

**Figure 6:** Some examples where concept algebra is not preferred. Each row has style-reference (leftmost), a content-reference (2nd from left), and left-to-right (here, but randomly permuted in the survey) images generated from concept-algebra, composition and direct prompting.

### C.4 The nurse example (Figure 4)

We initiate the process with an original prompt, $x_{\text{orig}} = $ "a portrait of a nurse". Our goal is to perform concept projection to manipulate the sex attribute. Similar to the mathematician example, we define the sex direction using a pair of prompts: $(x_1, x_2) = $ ("a man", "a woman"). However, instead of setting the distribution of sex as one of the delta functions or a fair one corresponding to the neutral prompt "a person", we wish to see what the concept's arithmetic average will define. Specifically, we take the sex direction of the average of a female nurse and a male nurse, calculated as $\frac{1}{2}s$(a female nurse)$+\frac{1}{2}s$(a male nurse"). It turns out the arithmetic mean realize the interpolation between two extremal sex points in the sex subspace, and the score function after concept projection returns images of androgynous nurses.

## D  Additional experiments

**Concept Algebra beats negative prompting in simple tasks**  Negative prompting, aims to eliminate target concept expressions by subtracting relevant scores. Unlike concept algebra and similar to concept composition, this method does not confine manipulations to specific subspaces. As predicted, we see negative prompting inadvertently modify off-target concepts tied to the primary concept, whereas concept algebra succeeds, as in .

**Concept algebra fails when causal separability (Theorem 3.2) is violated**  We show one concrete example of failures.

Figure 8 shows that we are unable to transfer the gender of the nurse when we calculate the score function of a male nurse by $(\mathbb{I}-\text{proj}_Z)s$["a portrait of a nurse"]$+\text{proj}_Z s$["a man"] where $\text{proj}_Z$ is computed by $s$["a buck on the grass"]$-s$["a doe on the grass"]. The target concept $Z$ and $W$ are sex $\in \{$male, female$\}$ and species$\in \{$human, deer$\}$. It's obvious that the sex and species have an interaction effect on the image $Y$ — different species induce different sexual characteristics.

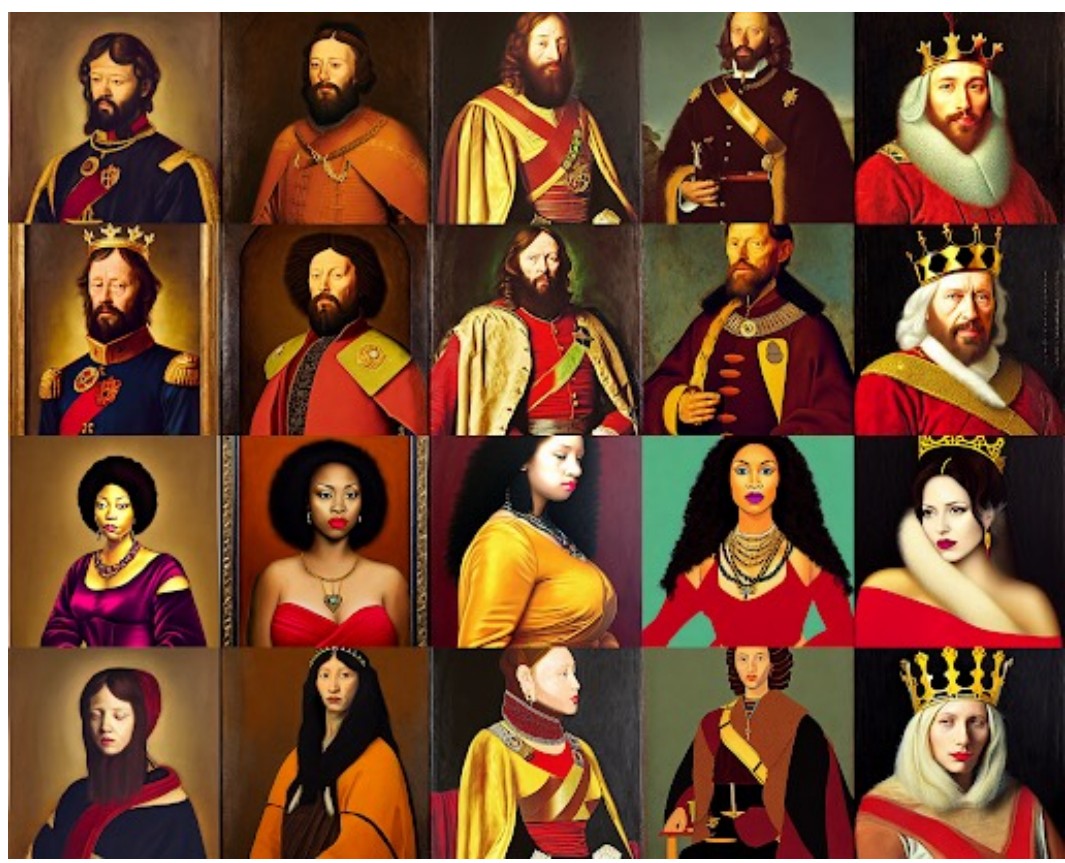

**Figure 7:** Concept-algebra can succeed when negative prompting fails. Images (matched on random seed) from top to bottom: direct prompting "a portrait of a king", negative prompting w/ "male", negative prompting w/ "male" but using concept projection, concept algebra with prompt "female". Note negative prompting does not remove the maleness of the output.

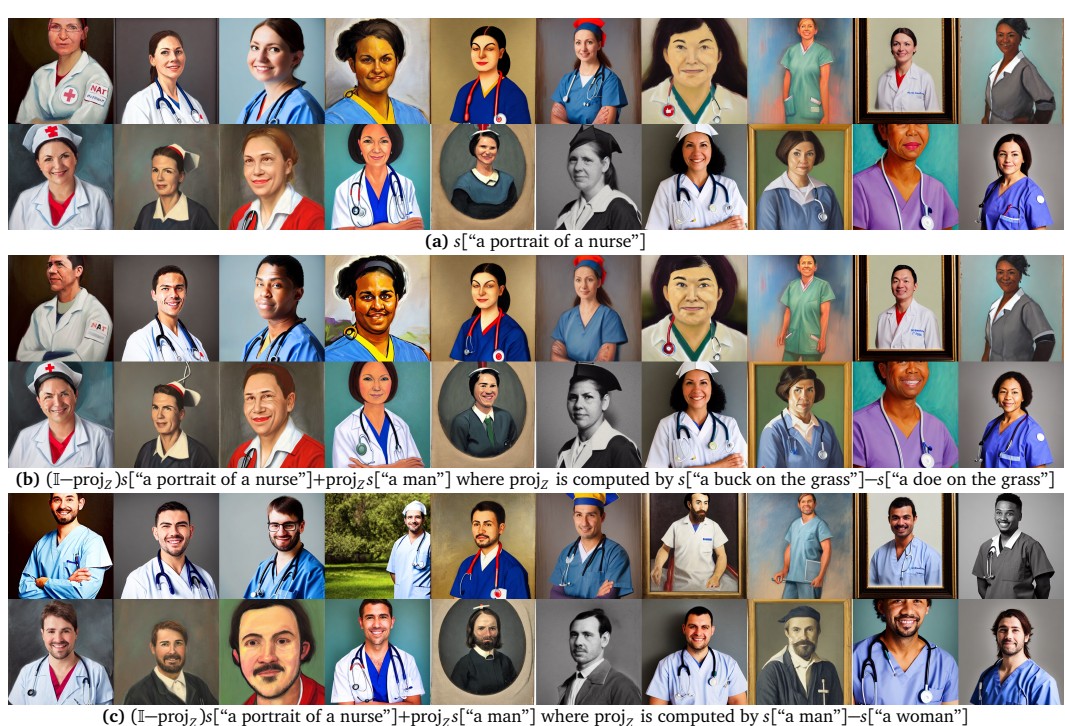

**(a)** $s[$"a portrait of a nurse"$]$

**(b)** $(\mathbb{I}-\text{proj}_Z)s[$"a portrait of a nurse"$]+\text{proj}_Z s[$"a man"$]$ where $\text{proj}_Z$ is computed by $s[$"a buck on the grass"$]-s[$"a doe on the grass"$]$

**(c)** $(\mathbb{I}-\text{proj}_Z)s[$"a portrait of a nurse"$]+\text{proj}_Z s[$"a man"$]$ where $\text{proj}_Z$ is computed by $s[$"a man"$]-s[$"a woman"$]$

**Figure 8:** Necessity of Assumptions: the validity of concept algebra depends on causal separability

