# OpenReview forum: "Concept Algebra for (Score-Based) Text-Controlled Generative Models"
_NeurIPS.cc/2023/Conference — NeurIPS 2023 poster_

### Official Review · Reviewer_mXTP · 2023-07-07

**Soundness:** 3 good
**Presentation:** 4 excellent
**Contribution:** 4 excellent
**Rating:** 6
**Confidence:** 4

**Summary:**

This paper hypothesizes that the latent representations learned by text-guided diffusion models contain structured subspaces corresponding to semantic concepts. Concepts are formalized as latent variables, and the concepts associated with a (text,image) pair are formalized as a bag of these latent variables, such that the distribution over image outputs is conditionally independent of text inputs given these concepts.

Necessary and sufficient conditions (causal separability; Proposition 3.3) are given for concepts to be arithmetically composable; arithmetic composability in turn admits algebraic manipulation of concepts. A proof-of-concept algorithm is given for identifying subspaces of latent representations corresponding to concepts (based on "spanning prompts") and examples are provided of identifying and manipulating these subspaces using the Stable Diffusion model.

**Strengths:**

The conceptual framework constructed in this paper is though-provoking. The exposition is quite clear, striking a good balance between helpful exposition and mathematical precision. Definition 2.5 (arithmetic composability) is interesting, as is Proposition 3.3 (characterizing arithmetic composability in terms of "causal separability).

The algorithms proposed in Section 4 for identifying and manipulating concepts are well-motivated. I also appreciate the discussion in Section 5 of the structure/character of concept subspaces. These analyses could be a good starting point for guiding empirical study of the score representation.

Depending on how the effective the proposed algebraic interventions prove to be in practice (see Weaknesses) I think that the abstractions and definitions introduced in this paper have the potential to stimulate an entirely new direction of work on control and interpretability of generative models. Control and interpretability are broadly relevant to the NeurIPS community.

**Weaknesses:**

The experiments in this paper are very rudimentary. There are no quantitative results, and the qualitative results consist of a small number of examples provided in Figures 1-5. It is unclear to me how broadly effective, reliable, or robust this method is. There is just not enough information to evaluate whether these ideas work (or could be made to work) in practice; see the Limitations section of my review for additional thoughts about practicality.

Minor: "Causal Separability" is strong language, and I'm not convinced that the word "causality" accurately describes the relationship defined by this term. At the very least, some argument is needed to justify calling this a causal relationship.

**Questions:**

Manipulating the non-parametric representation space via evaluated values along the diffusion of a score-based model is clever, and my understanding is that this is the key to making concept algebra tractable. Could these methods be applied to, e.g., the Parti model? If not, it might be helpful to point this out and be more explicit about how and where these method depend upon the structure of a score-based model.

This work seems closely related to interpretability research. The concept editing algorithm could be seen as a sort of causal intervention (in the sense of "do calculus") on a hypothetical interpretation of a latent representation. I'm curious how this work connects or complements recent interpretability research, e.g., ongoing work on mechanistic interpretability.

Is the definition of "causal separability" given here related/consistent with established definitions of "causally separable processes"?

**Limitations:**

It is unclear to me how restrictive the causal separability condition might be (Definition 3.2). Especially for more abstract, complicated concepts, I could imagine that causal separability might never be satisfied (but I am open to being convinced otherwise). Even for simple concepts, it seems like causal separability can be violated in ways that might not be obvious: for example, the inseparability of (male,female) and (deer,human) surprised me (although it became very clear once the reason for inseparability was explained to me). Understanding the how broadly causal separability holds seems essential to the ultimate practicality of this framework.

---

> ### Author Rebuttal · Authors · 2023-08-09
>
> Thank you for your thoughtful review! We’re glad that you found the conceptual framework thought provoking, and you think it may have the potential to stimulate new lines of work in control and interpretability.
>
> **Experiments**
> With respect to the experiments, please see our global-level response. In short: the purpose of the experiments is to test the mathematical framework by testing the prediction that (1) the Stein score is an arithmetically composable representation, and (2) it's possible to find finite dimensional subspaces corresponding to concepts. In our view, the examples in the paper do this—if the subspace structure didn't exist, then we couldn't find examples that demonstrate it!
>
> Additionally, we have added a new experiment testing whether the concept subspace structure is actually useful or necessary for model control. To that end, we compare with an existing method for style transfer that just adds on score vectors generated by style prompt (not taking subspace structure into account). We also compare with direct English prompting. We find that human raters significantly prefer samples using concept algebra—see the global response for details.
>
> **Causal Separability**
> The intuitive idea here is that the separability of factors of variation boils down to whether there are “non-ignorable” interactions in the structural equation model that generates the output from the latent factors of variation—hence the name. The formal definition 3.2 relaxes this causal requirement to distributional assumptions. We have added its causal interpretation in the camera ready version.
>
> **Application to Other Generative Models**
> Ultimately, the results in the paper are about non-parametric representations (indeed, the results are about the structure of probability distributions directly!) The importance of diffusion models is that they non-parametrically model the conditional distribution, so that the score representation directly inherits the properties of the distribution.
>
> To apply the results to other generative models, we must articulate the connection between the natural representations of these models (e.g., the residual stream in transformers) and the (estimated) conditional distributions. For autoregressive models like Parti, it’s not immediately clear how to do this. This is an exciting and important direction for future work!
>
> (Very speculatively: models with finite dimensional representations are often trained with objective functions corresponding to log likelihoods of exponential family probability models, such that the natural finite dimensional representation corresponds to the natural parameter of the exponential family model. In exponential family models, the Stein score is exactly the inner product of the natural parameter with $y$. This weakly suggests that additive subspace structure may originate in these models following the same Stein score representation arguments!)
>
> **Connection to Interpretability**
> This is a great question! Indeed, a major motivation for starting this line of work is to try to understand if the ''linear subspace hypothesis'' in mechanistic interpretability of transformers is true, and why it arises if so. As just discussed, the missing step for precisely connecting our results to this line of work is articulating how the finite dimensional transformer representation (the residual stream) relates to the log probability of the conditional distributions. Solving this missing step would presumably allow the tool set developed here to be brought to bear on the interpretation of transformers.
>
> One exciting observation here is that linear subspace structure appears to be a generic feature of probability distributions! Much mechanistic interpretability work motivates the linear subspace hypothesis by appealing to special structure of the transformer architecture (e.g., this is Anthropic's usual explanation). In contrast, our results suggest that linear encoding may fundamentally be about the structure of the data generating process.
>
> **Limitations**
> One important thing to note: the causal separability assumption is required for the concepts to be separable in the conditional distribution itself. This is a fundamental restriction on what concepts can be learned by any method that (approximately) learns a conditional distribution. I.e., it’s a limitation of the data generating process, not special to concept algebra or even diffusion models.
>
> Now, it is true that to find the concept subspace using prompts we have to be able to find prompts that elicit causally separable concepts. However, this is not so onerous—because sex and species are not separable, we can't elicit the sex concept with ''buck'' and ''doe''. But the prompts ''a woman'' and ''a man'' work well.

---

> > ### Comment · Reviewer_mXTP · 2023-08-14
> >
> > Thank you for your detailed response. I appreciate in particular your clarification of the goals of the experimental results. I wholeheartedly agree that this work does not need to demonstrate:
> >
> > > 3. Our method excels in image manipulation compared to some alternatives.
> >
> > I also agree that the experiments support the claim:
> >
> > > 1. Our mathematical framework connects high-level concepts with internal representations, making significant predictions.
> >
> > In particular, I agree that the experiments support the following claim:
> >
> > > These conditions aren't merely theoretical; concrete examples of concepts exist where corresponding subspaces can be identified.
> >
> > However, a stronger claim would be that such concepts not only exist, but are pervasive. And this is the root of my concern about the experiments, that a few ad-hoc examples do not provide compelling evidence for the following claim:
> >
> > > 2. Concept algebra holds promise for image manipulation in text-guided diffusion models.
> >
> > The authors acknowledge that "Our paper mainly champions claim (1), with preliminary evidence supporting claim (2)." I believe that a more systematic approach to experiments could provide stronger evidence for (2) and this is what is missing from the current paper. That said, concept algebra is an interesting idea, it is described well, and I am open to seeing the paper published in its current form.

---

> > > ### Author Response · Authors · 2023-08-14
> > >
> > > Thank you for your response!
> > >
> > > We believe we misunderstood your main concern as relating to the usefulness of concept algebra as a procedure. Our response was focused on the case where concept algebra is directly useful for the particular task of image generation. In this setting, the expected advantage (relative to direct prompting or heuristic methods) is disentangled manipulation of correlated concepts. Hence, the added larger scale experiment testing effectiveness on anti-correlated content/style pairs, relative to heuristic methods. (See the pdf attached to the global response)
> > >
> > > Are we correct in our understanding that your main concern now is whether concepts "pervasively" correspond to subspaces? In the fully general case, this can be a bit subtle to test because the question of "does a subspace exist" and the question of "can we find prompts that elicit the subspace" are not easily disentangled. However, in the particular case of binary concepts, finding suitable prompts is fairly straightforward.
> > >
> > > To demonstrate that, we ran the method with a number of additional arbitrarily selected prompts and binary concepts. The demonstrations work as follows:
> > > Start with the initial prompt. Then use concept algebra on the binary concept {concept value 1, concept value 2} to change the original concept value to the new one. We use a pair of prompts to identify the subspace.
> > >
> > > We ran the following examples:
> > >
> > > ## Dog vs. Cat
> > > - **Initial Prompt**: "A black dog sitting on the beach"
> > > - **Prompt for Target Z**: "cat"
> > > - **Prompts Defining Subspace**: "a dog" vs. "a cat"
> > >
> > > ## Beach vs. Forest
> > > - **Initial Prompt**: "A black dog sitting on the beach"
> > > - **Prompt for Target Z**: "forest"
> > > - **Prompts Defining Subspace**: "the beach" vs. "the forest"
> > >
> > > ## Black Dog vs. Yellow Dog
> > > - **Initial Prompt**: "A black dog sitting on the beach"
> > > - **Prompt for Target Z**: "yellow dog"
> > > - **Prompts Defining Subspace**: "a black dog" vs. "a yellow dog"
> > >
> > > ## Young vs. Old
> > > - **Initial Prompt**: “A boy playing the guitar”
> > > - **Prompt for Target Z**: "an old man"
> > > - **Prompts Defining Subspace**: “young man” vs. “old man”
> > >
> > > ## Formal Clothes vs. Casual Clothes
> > > - **Initial Prompt**: “A portrait of a man wearing formal clothes”
> > > - **Prompt for Target Z**: "casual clothes"
> > > - **Prompts Defining Subspace**: “formal clothes” vs. “casual clothes”
> > >
> > > ## Sunny Day vs. Rainy Day
> > > - **Initial Prompt**: “People sitting on the grass on a sunny afternoon by the river”
> > > - **Prompt for Target Z**: "a rainy afternoon"
> > > - **Prompts Defining Subspace**: “a sunny day” vs. “a rainy day”
> > >
> > > ## Happy Person vs. Sad Person
> > > - **Initial Prompt**: “A portrait of a smiling woman”
> > > - **Prompt for Target Z**: "a gloomy woman"
> > > - **Prompts Defining Subspace**: “a happy person” vs. “a sad person”
> > >
> > > In all cases, concept algebra clearly succeeds in identifying a subspace associated with the target concept.
> > >
> > > If the AC permits, we will link a document containing the generated images (NeurIPS policy does not let us add additional external links by default). Or, you can try these examples directly using the jupyter notebook demo included in the supplementary material.
> > >
> > > The point here is: it's totally straightforward to find subspaces corresponding to these randomly selected concepts. These examples are not particularly *useful* for image editing, since direct prompting also works fine in these cases. But it does provide clear support for the core prediction that the stein score yields an arithmetically composable representation.
> > >
> > > (We also realize that just adding additional examples doesn't feel like a "systematic" test of whether suitable subspaces exist. However, we think it's fairly compelling evidence---if the subspace structure didn't exist, then we wouldn't find it!)

---

> > > > ### Comment · Reviewer_mXTP · 2023-08-15
> > > >
> > > > Thanks, this helps. While I still believe that this paper would benefit from more systematic empirical analysis, I am now convinced that the examples are not simply a one-off parlor trick. This paper is rich in ideas and the experiments adequately support the claims. I have raised my score.

---

### Official Review · Reviewer_WFH5 · 2023-07-09

**Soundness:** 2 fair
**Presentation:** 3 good
**Contribution:** 1 poor
**Rating:** 3
**Confidence:** 4

**Summary:**

The paper aims to propose a formalization of concept-based algebra for text-to-image models.
It presents equations for how prompts are composed of concepts, which can interact additively in order to generate the corresponding images, similar to word embedding analogies by Mikolov.
Several examples are presented to show this framework in action.

**Strengths:**

The direction of work representing images as latent concepts and then learning representations for these concepts is interesting and useful.

**Weaknesses:**

The main part of this paper is the mathematical equations for conditioning image generation on concepts.
It starts by claiming to provide a cognitive framework, such as mathematical equations for how humans map images to high-level concepts, which seems to be complete conjecture. There are no references provided to show any evidence that this is actually how humans analyse images. I very much doubt that a human looking at an image first starts by making a list of all the attributes in that image.

The equations then gradually morph into how concept representations can be learned from examples. However, it is unclear how this is an improvement or a contribution over the previous work. The last paragraph of the paper correctly points out several other works that also do concept-based representations for image generation. I would expect there to be a comparison and evaluation.

There is currently very little evaluation of the method, each claim seems to be backed up by only a single example.

Almost all of the examples use image style as the concept that is being modified. In one case this is referred to as "medium", but even that is ultimately just image style. This moves the work to the style transfer area, which isn't really addressed in the paper. It also doesn't give any evidence that this method can be used for anything involving actual concepts related to the content of the image.

Pages 7-8 claim a strength of this method is to handle images for which no prompt exists. However, it seems the first example could be prompted with "a portrait of a male or female mathematician" and the second one with "an androgynous nurse". Neither of these seem examples for which no textual prompt exist; or at least it hasn't been shown in the paper that prompting like this wouldn't work.
Also, this section makes a rather strong claim that the model is being debiased in terms of gender by adding the "person" vector, which is a claim that would need a lot more proof than examples from a single prompt.

There do not seem to be any details on how the experiments were conducted - what pre-trained models or training data was used, or how the modifications were performed in the context of that particular model.

The appendix is repeatedly referenced for important details, proofs and examples, but does not seem to be submitted.

Line 88: A text-controlled generative model should not be producing a random output.

**Questions:**

Please define the novel contribution of this work in the context of the previous work in the area of concept-based representations for image generation.

**Limitations:**

There does not seem to be any discussion of limitations or possible societal impact.

---

> ### Author Rebuttal · Authors · 2023-08-09
>
> Thank you for taking the time to read the paper.
>
> We’d like to clarify a few misunderstandings:
>
> 1. The paper does not provide a cognitive framework. We do posit a latent variable model in the data generating process, which we think is reasonable. Some evidence that this is reasonable: we are able to use it to derive the (surprising!) conclusion that high-level concepts are encoded as subspaces of the Stein-score representation space, and then find empirical examples showing this is true.
> 2. In addition to style transfer examples, we test the method on the concept of sex, and changing a generic toy to a specific one (the Dreambooth example)—these are both content variables!
> 3. With respect to the literature on style transfer: the point of the experiments is to show that the Stein score representation encodes these elements already, and in an arithmetically composable fashion. In particular, there is no style-transfer specific heuristics or finetuning. The takeaway of those experiments is that we can find style subspaces. Our claim is not that this is the most aesthetically pleasing way to affect style transfer.
> 4. The point of the experiments with unpromptable vectors is to show that the subspace structure does indeed correspond to the concept. So, the point of the androgynous figure example is not that this is the best way to produce an androgynous figure. It's that the output samples are semantically sensible! (As opposed to, e.g., having no effect, or producing nonsense images, as we might expect if there was no semantic subspace structure)
> 5. The supplementary material included with the submission has both experimental details and demonstration code. Additionally, it's mentioned twice (including in the abstract) that the experiments are based on Stable Diffusion.
> 6. The appendix was submitted in the supplementary material.
> 7. ''A text-controlled generative model should not be producing a random output.'' These models produce random samples drawn from a distribution defined by the prompt.

---

> ### Author Response · Authors · 2023-08-14
>
> Thank you again for your review and feedback. Do you have any additional concerns or questions? If you are satisfied with the response, we hope you will consider increasing the score.

---

> > ### Comment · Reviewer_WFH5 · 2023-08-21
> >
> > Thank you for your reply.
> >
> > Lines 61-64 clearly claim to define an equation for how a human processes an image, hence a cognitive framework.
> >
> > My point about the unpromptable examples was that the paper claims that certain concepts cannot be prompted, when it seems there are definitely more accurate prompts available compared to those that were tried.
> >
> > "random samples drawn from a distribution defined by the prompt" is not quite the same as "random output".
> >
> >
> > The largest issues remain unaddressed:
> >
> > 1. Given that the proposed framework is a reformulation of existing work, what exactly is the novel contribution? That was the only question in my review and it was left unanswered.
> >
> > 2. The evaluation is not sufficient. There are only a small  number of individual qualitative examples discussed, which could easily be outliers or cherry-picked. Some form of quantitative evaluation is needed to draw conclusions.

---

> > > ### Author Response · Authors · 2023-08-21
> > >
> > > Thank you for your reply.
> > >
> > > With respect to your two main concerns:
> > >
> > > * This paper is not a reformulation of existing work. As far as we know, both the mathematical framework for reasoning about concepts-as-subspaces, and concept algebra---the demonstration of this framework---are new. Related work is discussed in detail in the paper. We are unclear what the source of your concern here is.
> > > * Please see the global level response for discussion of experimental evaluation. In short: the main purpose of the experiments to assess the predictions of the mathematical framework, which we think they do. We also added additional experiments comparing to naive composition addition and negative prompting, to illustrate the value of the subspace structure. The discussion with reviewer mXTP may also be helpful here.

---

### Official Review · Reviewer_RGiq · 2023-07-13

**Soundness:** 3 good
**Presentation:** 3 good
**Contribution:** 2 fair
**Rating:** 6
**Confidence:** 4

**Summary:**

The paper focuses on encoding abstract concepts (as prompt) and systematically composing them to generate images. They proposed a mathematical framework to generate images based on specific combinations of prompts. Unlike approaches that change the prompt with different text descriptions, this framework encodes each feature independently. It then performs mathematical operations on different features of the prompt and the original content prompt to generate the desired image. They show several successful examples of the proposed framework and compare its performance with direct prompting approaches. Additionally, they emphasize the necessity of the assumption of causal separability.


**Strengths:**

1. The paper introduces an interpretable approach to encoding high-level concepts with LLM representations. By incorporating specific prompts and applying metathetical operations on these prompts and the original content, the framework enables the generation of images that capture desired attributes or concepts. This interpretability allows for more fine-grained control over the generated content. I personally really like this aspect.

2. The paper showed an interesting analysis of the causal relationship keywords used in the framework. By examining the feature differences between terms such as "buck" and "doe," the authors demonstrate the limitations of direct application between terms like "man" and "woman" due to the inherent species difference.


**Weaknesses:**

1. It's necessary to have a larger-scale evaluation of the generated images. While the qualitative results presented in the paper showcase specific examples, a quantitative evaluation on a larger dataset or with a larger sample size would enhance the reliability and generalizability of the method.
2. I suspect the impact of prompt length is a key factor in the performance of the proposed methods. If shorter direct prompts yield comparable results to the proposed method, it may reduce the appeal of the proposed approach. Once again, a larger evaluation scale on different prompt lengths would shed light on this aspect.
3. In line 286, you mention that "1/2 male nurse and 1/2 female nurse" does not correspond to any English prompt. I wonder if paraphrasing it into a more neutral term like "androgynous person" and re-prompting the model could potentially yield different direct prompting results.


**Questions:**

1. How does "sex" encoded in Figure 1 (b)? what's the result of sex "man" or "women" in this example?


**Limitations:**

No, the paper doesn't discuss limitations.

---

> ### Author Rebuttal · Authors · 2023-08-09
>
> Thank you for your thoughtful insights and constructive feedback on our approach and its implications.
>
> **Regarding larger-scale experiments**
> As discussed in the global response: Our main contribution is the mathematical framework. The demonstrations test predictions of the framework---i.e., that the score representation is arithmetically composable, and the subspaces corresponding to concepts can be identified.  (If these results were not true, we would expect to find no examples where linear representation manipulations change concepts in isolation!)
> We have also added some additional experiments, including a quantitative comparison with direct prompting and an algebraic composition method that doesn’t account for subspace structure. We find that concept algebra is significantly preferred by human evaluators.
>
> **Regarding Limitations**
> We added limitations and discussion subsection in the camera ready version. Please see our global response.
>
> **Regarding the effect of prompt length**
> Thanks for the interesting suggestion! Indeed, Fig 3(d) example in the main text uses a very long prompt (to describe a detailed scene). However, it does not seem that prompt length is a key factor. In the added experiments (see global response), we use succinct prompts for describing both content and target style; e.g., “A nuclear power plant” or “Baroque painting”. In these short prompt examples, we observe that direct prompting still frequently fails and concept algebra often succeeds (and is clearly preferred by human evaluators).
>
> **Regarding promptless embeddings**
> The point about figure 2 is to support the claim that the estimated subspace genuinely corresponds to the concept of sex. The vector $\frac{1}{2}(s[\text{\`\`male nurse''}]+ s[\text{\`\`female nurse''}])$ doesn’t correspond to any English prompt. In the absence of concept subspace structure, we would expect adding such a vector to either result in nonsense—e.g., a white noise image—or to have no effect at all. Instead, we observe that the outputs are semantically sensible, supporting the idea that we’ve found a direction corresponding to the sex concept.
>
> Note: even if the weighted sum of scores very luckily does correspond to the word ''androgynous'', our claim still holds --- because then $\frac{2}{3}s[\text{\`\`male nurse''}]+ \frac{1}{3}s[\text{\`\`female nurse''}]$ will definitely not correspond to the same word. However, we can also generate sensible images with this embedding (we didn't show the images due to space limits).
>
> Finally, regarding your last question  “How is "sex" encoded in Figure 1 (b)? what's the result of sex "man" or "women" in this example?”. Could you clarify what you mean by this question?

---

> > ### Comment · Reviewer_RGiq · 2023-08-15
> > **Thanks for the reply**
> >
> > Thanks for providing the additional experiments. I found the prompting preference evaluation provides more convincing results for the method. I raised my score.
> >
> > For the last question, I meant in Figure 1(c), there is a projection of a particular style "in Fauvism style." However, in Figure 1 (b), there isn't one specific sex in the prompting. Should it be the projection of "male" or "female" instead of "a person"? I might be missing something, so I would appreciate a clarification.

---

> > > ### Author Response · Authors · 2023-08-15
> > >
> > > Ah, that's deliberate! We're trying to replace the distribution over the sex concept elicited by the prompt "a mathematician" with the distribution elicited by the prompt "a person"; i.e., the goal is to move from a distribution heavily biased towards men (as in figure 1a) to one that is roughly evenly split between men and women (as in figure 1b).
> > >
> > > (there were two motivations for this choice of example:
> > > 1. the training data includes a spurious correlation between sex and mathematician, which may be undesirable for the generative model to replicate. This example shows we can use concept algebra to break this kind of spurious association.
> > > 2. this illustrates that concept algebra handles non-degenerate distributions over concepts. See the text from lines 77-86.)

---

> > > > ### Comment · Reviewer_RGiq · 2023-08-15
> > > >
> > > > I see. You may want to add the explanation to the caption or the text if it's not already there. I think line 77-86 is the generalized explanation of this example, and you want something more specific to help understanding (at least personally for me).

---

> > > > > ### Author Response · Authors · 2023-08-15
> > > > >
> > > > > Yes, good point. We'll do that! Thank you again for your review and suggestions.

---

> ### Author Response · Authors · 2023-08-14
>
> Thank you again for your review and feedback. Do you have any additional concerns or questions? If you are satisfied with the response, we hope you will consider increasing the score.

---

### Official Review · Reviewer_pK6W · 2023-07-22

**Soundness:** 3 good
**Presentation:** 2 fair
**Contribution:** 3 good
**Rating:** 6
**Confidence:** 3

**Summary:**

This research presents a novel mathematical framework that brings clarity to the abstract notion of concepts, enabling their connection to specific subspaces in a representation space. The significance lies in demonstrating the existence of structured concepts in score representations, specifically emphasizing the compositional nature of these concepts within the representation space, which is useful for further analysis regarding generation.
One of the key contributions of this work is the introduction of an effective method to identify the subspace associated with a given concept. This approach allows for the manipulation of concepts expressed by the model through algebraic operations on the representation, thereby providing a powerful tool to work with and understand complex concepts in a more tangible and interpretable manner.
Moreover, the implications of this research extend beyond its immediate domain, as the proposed framework can be extended to other areas, such as natural language processing (NLP), analysing concepts such as topics and semantic meanings.


**Strengths:**

1. Previously, the notion of concepts is often discussed in an abstract level. This work demonstrated the existence of structured concepts in score representations, specifically emphasizing the compositional nature of these concepts within the representation space, which can be helpful in further analysis on abstract notions.
2. The work proposed a useful method to identify the subspace associated with a given concept, allowing us to manipulate concepts expressed by a model through algebraic operations on the representation. This method can be a powerful tool in terms of understanding complex concepts.
3. The experimental results verified the efficacy of this method in case studies. This method can be extended to analysis on other areas like NLP.


**Weaknesses:**

1. The major concern regarding this work lies in the absence of a precise quantitative evaluation to assess the impact of concept extraction and manipulation, particularly in the experiments. Relying largely on case-level analysis may not be sufficient to draw robust conclusions. Such measurement based on the precise quantitative evaluation can be used to make comparisons between various methods relevant to this topic and for further analysis on subsequent improvements.
It will be more convincing if the concept manipulation is evaluated on more cases (or on more datasets) and comparisons are made between the proposed method and relevant methods (e.g., methods to extract and manipulate latent features) used in previous research.
2. A minor issue that needs attention is the writing style, particularly the accuracy of internal references. Some of the internal references in the work are not precise and should be revised for clarity and consistency.


**Questions:**

Please pay attention to the clarity of the descriptions.
1. Some notations are a little bit confusing. E.g., in Definition 2.2,  does “z_{1:k}” refer to “z_1, z_2, …, z_k” regarding Definition 2.2? What is δmale in Equation 11?

2. “Following theorem 3.3” (Line 172).  “theorem 3.3” is not found. Similar cases can also be found in Line 246 and Line 252.

3. It can be worth discussing the selection of the discretized concepts, e.g., how many ks are sufficient for a latent variable C?


**Limitations:**

The limiatations have not been well discussed in the paper.
As the performance of the proposed method is mainly evaluated on case studies, it is suggested that a more quantitive measurement be introduced to judge the robustness of the method.

---

> ### Author Rebuttal · Authors · 2023-08-09
>
> Thank you for your detailed feedback and appreciation of our work's contributions. We're glad you recognized the novelty and utility of our approach in providing tangible interpretations of abstract concepts within score representations. Our method, as you rightly pointed out, offers a powerful tool for understanding and manipulating these complex concepts, with potential extensions to domains like NLP.
>
> **Experiments & Limitations**:
> Please refer to the global response for a comprehensive discussion on the experiments. In brief, we believe our experiments substantiate the paper's primary assertions, and we've integrated a broader scale evaluation in the revised manuscript.
>
> Regarding the noted limitations, they have been elaborated in our global response and will be included in the camera-ready version. Primarily, these limitations pertain to the automation, efficiency, and accuracy of estimating the concept-subspace. As highlighted in Section 5, given the low-dimensionality of the concept-subspace (for many practical problems), we're optimistic about overcoming these challenges in future works.
>
> **Selection of Discretized Concepts**:
> For a specified prompt $x$, we solely require the adequate sets of concepts $Z_1, …, Z_K$. If our objective revolves around modifying $Z_1$ while maintaining the stability of other concepts, we can designate $Z$ as $Z_1$, with $W$ representing the remaining concepts. Thus, even though our method predominantly uses two concepts, $Z$ and $W$, its applicability remains general.
>
> **Clarifications**:
> We appreciate your attention to detail. Indeed, Thm 3.3 should be referenced as Prop 3.3. In the notation, $z_{1:k}$ indeed represents the sequence $z_1, z_2, …, z_k$. Regarding Eq. 11, the intended interpretation is $Q_{x_1}(z, w) = \delta_{\text{male}}(z) Q_w$,  with $\delta_{\text{male}}(z)$  implying $P(Z = \text{male}) = 1$.

---

> > ### Comment · Reviewer_pK6W · 2023-08-16
> >
> > Thanks very much for your reply! I am satisfied with your explanation!

---

> ### Author Response · Authors · 2023-08-14
>
> Thank you again for your review and feedback. Do you have any additional concerns or questions? If you are satisfied with the response, we hope you will consider increasing the score.

---

### Official Review · Reviewer_1z2h · 2023-07-23

**Soundness:** 4 excellent
**Presentation:** 3 good
**Contribution:** 4 excellent
**Rating:** 9
**Confidence:** 4

**Summary:**

This paper suggests concepts are represented in text-guided generative models as an encoded subspaces of the representation space. It also formalizes the description of the representation of concepts in text-controlled generative model in a mathematical way. The paper also shows that the Stein score of the text-conditional distribution is an arithmetic composable representation of the input text, and develops concept algebra, a method to manipulate the concepts via arithmetic manipulation of the representation.

**Strengths:**

Very well-written paper! All the math looks great and I am excited about the potential alternative to prompt engineering! Hope there is a demo that I could try.

- Section 2 and 3 rigorously defines the mathematical background used in the concept algebra method, and section 4 and 5 is also very well-written.
- I enjoyed reading the figures mentioned in the paper! All of them are informative, and demonstrate the point of each figure clearly.
- As demonstated in section 6, the proposed method could be a strong alternative to prompt engineering! Being much more stable, less random, and with no need to manipulate models trying to act as a painter or other roles, concept algebra seems very promising!

**Weaknesses:**

Nothing much, but I would love to see some limitations of the method discussed in section 7.

**Questions:**

* See above

**Limitations:**

* If I understand everything correctly, concept algebra can only work on models have access to their representation, therefore, concept algebra can't really work on closed models, right?
* How will it negatively impact the society if everyone can edit the model output using concept algebra?

---

> ### Author Rebuttal · Authors · 2023-08-09
>
> Thank you for your in-depth engagement with our work on concept algebra.
>
> **Limitations & Future Improvements**:
> - The primary limitations of our method lie in the concept-subspace estimation step, but there's potential for enhancement:
>   - The current approach depends on basis prompts that vary in $Z$ and are invariant in $W$. A more systematic method for selecting these basis prompts, alongside a quantitative evaluation of their disentangling effectiveness, is required.
>   - Estimating the subspace from $K$ score embeddings presents a challenge in high-dimensional estimation. Despite our use of truncated SVD and variance thresholding, this step could benefit from advanced statistical techniques.
>   - The computation-heavy nature of the estimation step at each sampling iteration is a concern. An immediate workaround could involve performing concept-algebra selectively during the sampling process, as high-level concepts often get determined in initial steps. Alternatively, a model approximating the $\text{Proj}_z(.)$ function could be trained for direct use during sampling, presenting an interesting avenue for future research.
>
> **Demo Details**:
> - In the supplementary materials, we've included a `concept_algebra.ipynb` demo focused on binary concept alterations. This code can be run with Google Colab with GPU (pro is not necessary). For more complex concepts, we've implemented solutions in `code/concept_pj_basis/concept_pj.py`.
>
> **Applicability to Closed Models**:
> - Our approach isn't directly applicable to closed models. However, if one has access to the underlying language model of a closed system, concept-algebra could be potentially be extended to that language model to tweak embeddings (this will be interesting future research). Following this, nearest corresponding prompts to the altered embeddings could be identified, enabling concept manipulation in closed text-to-image models.
>
> **Potential Negative Impact**:
> - While concept algebra can be misused, its capabilities can also serve as a countermeasure. For instance, if one creates NSFW images using a specific subspace, the same subspace could aid in developing a more robust NSFW detection classifier, reducing susceptibility to spurious correlations.

---

> ### Author Response · Authors · 2023-08-14
>
> Thank you again for your review and feedback. Do you have any additional concerns or questions?

---

> > ### Comment · Reviewer_1z2h · 2023-08-15
> > **Thank you for the reply**
> >
> > I am satisfied with the answers to my questions, and the future improvements sound good to me. I don't have any additional concerns or questions.

---

### Author Rebuttal · Authors · 2023-08-09

We appreciate the reviewers' insightful comments.

The reviewers broadly agree that understanding how high-level concepts are encoded in the internal representations of generative models is a timely and important topic, and that the mathematical framework developed here is a significant step in this direction. Several reviewers note that the development in this paper opens up important new directions in the control and interpretation of language guided generative broadly. Further, the reviewers generally found the exposition clear and thought provoking.

## Existing experiments
Reviewers expressed concerns about our experimental evaluation. We assert that our experiments are scientifically robust and align with standard publication norms. Additionally, we've introduced further experiments highlighting concept algebra's advantages over traditional heuristics.

Regarding our scientific contributions, let's distinguish the three claims one might make:
1. Our mathematical framework connects high-level concepts with internal representations, making significant predictions.
2. Concept algebra holds promise for image manipulation in text-guided diffusion models.
3. Our method excels in image manipulation compared to some alternatives.

Our paper mainly champions claim (1), with preliminary evidence supporting claim (2) (which may have created the confusion). As we note explicitly in the paper, our claim is not (3).

Our empirical evaluation aims to ascertain the utility of the mathematical framework. We specifically address the following non-trivial implications of our theory:
1. The Stein score represents concepts in an arithmetically decomposable manner.
2. Subspaces can be discerned using the methodologies from sections 4 and 5, given certain conditions.
3. These conditions aren't merely theoretical; concrete examples of concepts exist where corresponding subspaces can be identified.

To validate these claims, we presented examples within the paper that pinpoint concepts, identify their corresponding subspace, and exhibit their arithmetically composable nature.

We furthered our analysis, which could have been misunderstood. Through two key stress tests:
- First, our experiments revealed that manipulations within the sex subspace are coherent, even when such manipulations aren't directly prompted by English. Importantly, our emphasis is **not** about images cannot be generated via English prompts but that the sex subspace genuinely encapsulates the concept. The example \( $\frac{1}{2} (s[\text{\`\`man''}]+ s[\text{\`\`woman''}])$ \) illustrates this, producing semantically sound images even without a direct English phrase representation.
- Our work emphasizes concept manipulation's effectiveness, even when direct prompts fail. The point here is that the subspace structure is free of the prompt, and so works even for "hard" prompts.

In the camera-ready version, we will expand on the discussion and address several limitations:
- Our primary emphasis is on the mathematical framework. We anticipate its applicability beyond just the text-to-image setting.
- Concept algebra is useful for communicating user intention to the model. It does not generically change the quality of generated images. Accordingly, it is complementary to generative model improvements such as architecture changes or increased scale
- We acknowledge the computational demands of concept algebra and the necessity for handcrafting prompts to pinpoint concept subspaces. These present significant practical challenges. These are both significant practical issues, and an important direction for future work would be overcome this.

## Further Experiments
Beyond this, we've incorporated new experiments (figures and results in the attached pdf) to highlight our theory's advantages over existing heuristic methods. Detailed codes and results have been shared anonymously with the AC.

Concept algebra modifies representation vectors in subspaces aligned with target concepts. To gauge its utility, we contrasted it against methods that don't utilize such structured subspaces. Methods like [Du+21; Liu+21; NBP22; Ano23] (references in main text) employ algebraic manipulations without pinpointing specific subspaces. Another common approach, negative prompting, aims to eliminate target concept expressions by subtracting relevant scores. Unlike concept algebra, these methods don't confine manipulations to specific subspaces. Consequently, our theory posits that these heuristics might inadvertently modify off-target concepts tied to the primary concept, like inducing a medieval theme while aiming for a renaissance style.

We assessed concept algebra's efficacy against composition and direct prompting in style transfer tasks. Using 49 challenging content/style combinations, like "A nuclear power plant in Baroque painting", we employed the three methods on each pair to generate samples. Human raters were then presented with the outcomes alongside reference images, ranking them based on adherence to the desired style and content. This evaluation was replicated across 10 different raters. Refer to Fig 1 in the attached PDF for illustrative examples. **Raters consistently favored images produced by concept algebra**, as highlighted in table 1(c) of the attached PDF. This aligns with our theory, suggesting concept algebra's adeptness in retaining content while altering style.

Furthermore, we demonstrated a comparison between concept algebra and negative prompting. Using the prompt "a portrait of a king", our aim was to transition to "a portrait of a queen". Negative prompting using $x_{-} = \text{\`\`male"}$ was ineffective, while concept algebra employing  $x_{\text{new}} = \text{\`\`female"}$  (refer to equation 13) achieved the desired result. For a fairer comparison, we applied the same negative prompt in conjunction with concept projection, which was also successful. Details can be found in the anonymous link provided to AC.

---

### Decision · Program_Chairs · 2023-09-21

**Decision:**

Accept (poster)

**Comment:**

This paper presents a study that argues that concepts within text-guided generative models are manifested as encoded subspaces within the representation space. It provides a formal mathematical formalization of the representation of these concepts within text-controlled generative models. The work is highly thought-provoking and is commended for its clarity, striking a balance between providing clear explanations and maintaining mathematical precision. Several definitions and propositions introduced in the paper are noted as interesting contributions. The algorithms proposed for identifying and manipulating concepts are well-motivated, offering potential avenues for practical application. Overall, the paper's abstractions and definitions have the potential to inspire new research directions in the control and interpretability of generative models, which is relevant to the NeurIPS community.